

# Linking biodiversity and geodiversity: Arctic-nesting birds select refuges generated by permafrost degradation

Madeleine-Zoé Corbeil-Robitaille[1,2,3], Éliane Duchesne[2,3], Daniel Fortier[3,4], Christophe Kinnard[3,5], Joël Bêty[1,2,3]

[1]Department of Biology, Geography and Chemistry, University of Quebec at Rimouski, 300, allée des Ursulines, C.P. 3300, succ. A, Rimouski (Québec), Canada  G5L 3A1

[2]Canada Research Chair in Northern Biodiversity, 300, allée des Ursulines, C.P. 3300, succ. A, Rimouski (Québec), Canada G5L 3A1

[3]Center for Northern Studies, Laval University, Pavillon Abitibi-Price, Room 1202, Quebec City (Quebec), Canada, G1V 0A6

[4]Department of Geography in the Faculty of Arts and Sciences and Cold Regions Geomorphology and Geotechnics Laboratory, University of Montreal, Complexe des sciences, 1375 Avenue Thérèse-Lavoie-Roux
Montréal (Québec), Canada H2V 0B3

[5]Department of Environmental Sciences, University of Quebec at Trois-Rivières, 3351, boulevard des Forges, Trois-Rivières (Québec), Canada G8Z 4M3

*Correspondence to*:  Madeleine-Zoé Corbeil-Robitaille (mzoecr@gmail.com)

**Abstract.** To gain better insight into the cascading impact of warming-induced changes in the physical landscape on biodiversity, it is crucial to establish stronger links between abiotic and ecological processes governing species distribution. Abiotic processes shaping the physical characteristics of the environment could significantly influence predator movements in the landscape and ultimately affect biodiversity through interspecific interactions. In the Arctic tundra, the main terrestrial predator (Arctic fox) avoids patches of wetlands composed of ponds with islets that can act as refuges for prey. Little is known about the geomorphological processes generating islets selected by prey species. Our study aimed to identify i) the physical characteristics of islets selected by Arctic-nesting birds and ii) the geomorphological processes generating islets available in the landscape. Over two breeding seasons, we determined the occurrence of nesting birds (Glaucous gull, Cackling goose, Red-throated loon) on islets (N=396) found over a 150 km$^2$ area on Bylot Island (Nunavut, Canada). Occupied islets were located further away from the shore (10.6m ± 7.3 vs 7.4m ± 6.8) and surrounded by deeper water (33.6cm ± 10.6 vs 28.1cm ± 11.5). As expected, all three bird species selected islets less accessible to Arctic foxes, with nesting occurrence increasing (linearly or nonlinearly) with distance to shore and/or water depth around islets. Based on high-resolution satellite image and field observations, we found that ice-wedge polygon degradation generated the majority of islets (71%) found in the landscape. Those islets were on average farther from the shore and surrounded by deeper water than those generated by other processes. As polygon degradation is projected to accelerate in response to warming, new refuges will likely emerge in the Arctic landscape, but current refuges could also disappear. Changes in the rate of polygon degradation may thus affect Arctic tundra biodiversity by altering predator-prey interactions.



## 1    Introduction

The heterogeneity of the Earth's abiotic surface, referred to as geodiversity, is increasingly gaining recognition as a pivotal
force shaping the diversity of biological communities (Vernham et al., 2023; Schrodt et al., 2019). Geodiversity characterizes
the available physical environments and can shape species distribution (Burnett et al., 1998; Lawler et al., 2015). Therefore,
inclusion of geodiversity in biodiversity research can improve our understanding of biodiversity patterns and our ability to
anticipate the impact of climate changes on wildlife (Brazier et al., 2012; Tukiainen et al., 2022; Alahuhta et al., 2020). In this
context, it is imperative that we establish robust connections between key abiotic processes affecting the physical landscape
with ecological dynamics governing species interactions and distribution.

Nowadays, it is well established that the physical landscape of the Arctic tundra is strongly affected by global warming through
permafrost-related changes (Jorgenson et al., 2010; Liljedahl et al., 2016; Farquharson et al., 2019). Climate change is causing
deeper active layer development and thaw of permafrost in many Arctic regions (Bonnaventure and Lamoureux, 2013).
Climate projections predict higher air temperatures and increased precipitations, and model results indicate that the active layer
will likely deepen, and permafrost loss will continue (Shur and Jorgenson, 2007; Lawrence et al., 2008; Farquharson et al.,
2019). These changes can affect the surface stability, as well as surface drainage and ponding (Liljedahl et al., 2016; Lantz and
Kokelj, 2008), leading to potential alterations of habitats used by wildlife (Berteaux et al., 2017).

Predation is one of the key biotic interactions that can shape species distribution at various spatiotemporal scales (Lima, 1998;
Wisz et al., 2013; Menge and Sutherland, 1976). Physical characteristics of the environment can hinder predator movements
in a landscape (Caro; Timothy, 2005) and create habitat patches with reduced predation risk, which can be used by prey species
to avoid predation (i.e., prey refuges (Gauthier et al., 2015; Sih, 1987)). The presence of refuges in the landscape can contribute
to the persistence of species vulnerable to predation and partly drive spatial distribution patterns of both predators and prey
(Holt, 1987; Berryman et al., 2006; Lima, 1998).

In Arctic vertebrate communities, prey refuges can promote species occurrence and coexistence (Duchesne et al., 2021;
Clermont et al., 2021; Léandri-Breton and Bêty, 2020). For example, terrestrial predators like Arctic foxes tend to avoid patches
of wetlands composed of ponds with islets that can limit their movements compared to surrounding dryer habitats (Grenier-
Potvin et al., 2021). Islets can thus act as important refuges commonly used by tundra prey, such as Arctic-nesting birds
(Gauthier et al., 2015; Sittler et al., 2000; Lecomte et al., 2008). Species using islets can be less affected by spatial and annual
variation in predation pressure, and hence the presence of refuges like islets can modulate species interactions and distribution
in the landscape (Clermont et al., 2021; Duchesne et al., 2021). Anticipating the impact of warming on the availability of
refuges in the Arctic tundra is challenging due to our limited understanding of the abiotic processes that create the refuges
preferred by various Arctic prey species.





In this study, we aim to i) identify the physical characteristics (such as distance to shore, water depth, surface areas) of islets selected as refuges by Arctic-nesting birds and ii) identify the main geomorphological processes responsible for forming islets in a High-Arctic tundra landscape (see Fig.1). We first mapped and characterized the islets found on the southwest plain of
Bylot Island, located north of Baffin Island in the Canadian Arctic. We then examined whether islet characteristics affect nest site selection by three tundra bird species known to nest mostly on islets (Glaucous gull (*Larus hyperboreus*); Cackling goose (*Branta hutchinsii*); Red-throated loon (*Gavia stellata*)). We hypothesized that birds would select islets less easily accessible by Arctic foxes (i.e., those farther from the shore and surrounded by deeper water). Using satellite image and field observations, we further associated each islet with a specific geomorphological or biotic process underlying its presence in the landscape.
As surface hydrology, microtopography and permafrost dynamics strongly interact in the Arctic (Liljedahl et al., 2016; Woo and Young, 2003; Nitzbon et al., 2019; Khani et al., 2023), we expected that permafrost-related geomorphological processes would generate a large proportion of the islets available in flat lowlands and upland plateaus throughout the study area.

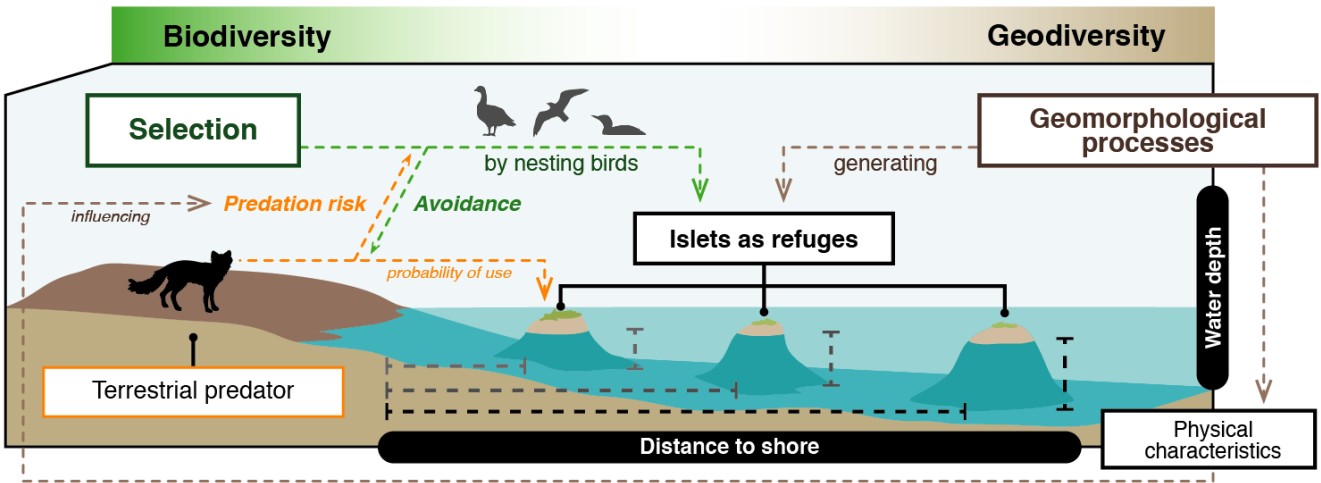

**Figure 1. Schematic representation of the link between geomorphological processes, physical characteristics of the landscape and predator-prey interaction in the Arctic tundra. Birds are expected to select islets less easily accessible by the main predator, the Arctic fox (those farther from the shore and surrounded by deeper water) because it may reduce nest predation risk. Hence, geomorphological processes that generate physical characteristics that hinder Arctic fox movements could influence Arctic birds' distribution pattern and abundance.**

**2    Methods**

**2.1    Study area**

We conducted summer fieldwork over two years (2018–2019) on the southwest plain of Bylot Island, a vast Migratory Bird Sanctuary in Sirmilik National Park, Nunavut, Canada (72°54′N, 79°54′W). The study area is composed of flat lowlands and



upland plateaus incised by valleys and glacial rivers, and characterized by extensive continuous permafrost (active layer ranges
between 30cm to 100cm deep (Fortier et al., 2006)). Most of the area is covered with mesic tundra in the uplands and an
assemblage of mesic tundra and wetlands in the lowlands (Gauthier et al. 2013). Lakes, ponds and complex of polygonal
wetlands are scattered across lowlands and coastal areas.

More than 35 bird species, including waterfowl, shorebirds, seabirds, raptors, and passerines nest in the study area (Lepage et
al. 1998). Three of these species nest essentially on small islets in water bodies (Fig.2): the Glaucous Gull *(Gulls)*; the Red-
throated Loon *(Loons)* and the Cackling Goose *(Geese)*. The Arctic fox *(Vulpes lagopus)*, a generalist predator, is the main
nest predator in our study system (Bêty et al., 2001; McKinnon and Bêty, 2009; Giroux et al., 2012). Avian predators, such as
glaucous gulls, ravens (*Corvus corax*) and jaegers *(Stercorarius parasitus)* do not represent the main cause of nest failure but
can nonetheless prey upon eggs of various bird species (Gauthier et al., 1996; McKinnon and Bêty, 2009).

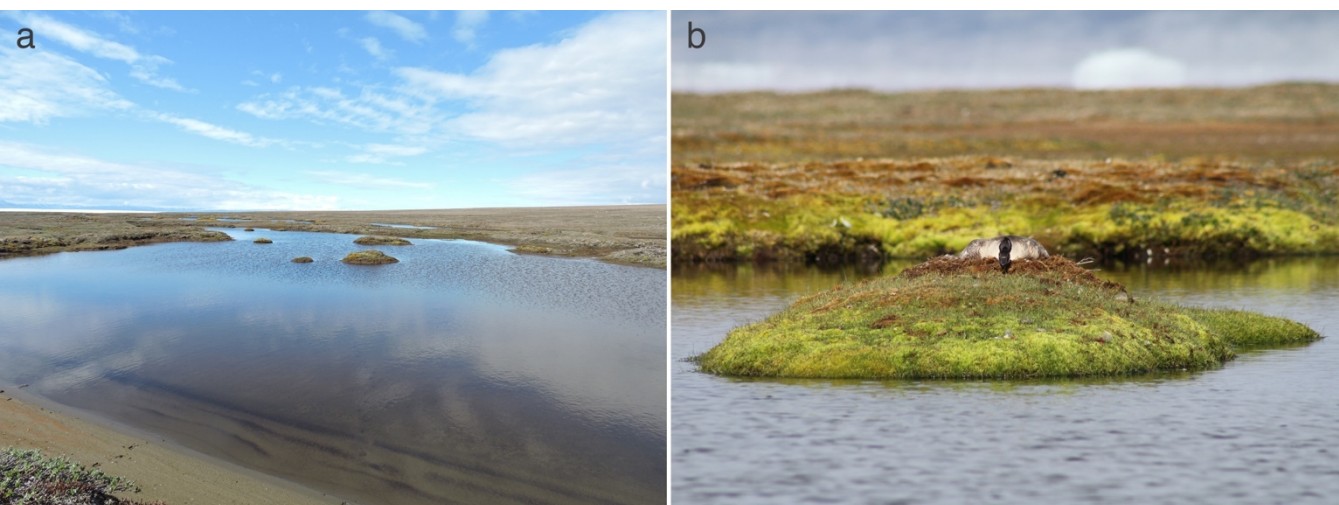


**Figure 2. Pictures illustrating a typical water body with few islets (a) and a cackling goose nesting on an islet (b) on Bylot Island
(Nunavut, Canada). Photo credits: Jeanne Clermont (a) and Yannick Seyer (b).**

**2.2    Islet characteristics and selection**

We georeferenced islets in the study area using a combination of satellite image analyses and intensive field surveys carried
during the bird incubation period (between late June and mid-July). We are confident that we found the vast majority of the
islets in the study area. We described islets using the following characteristics: 1) distance to shore (hereafter DISTANCE:
shortest distance in meters between the shore and the islet; measured on foot, ±1m) and 2) water depth (hereafter DEPTH:
maximum water depth in centimeters recorded on the shallowest, generally the shortest, route between the shore and the islet;
measured using a graduated walking stake, ±5cm). These two variables are the ones we aim to focus on, as we hypothesize
that these characteristics can impede Arctic fox movement (Strang, 1976). We also considered the islet area (hereafter
IsletArea; exposed surface of the islet in square meters) and the waterbody area (hereafter LakeArea; waterbody surface



entirely covered by water in square meters) as they could be additional proximal parameters used by birds to select their nesting site, whether because of physical restrictions, to nest, to raise young or to carry out their daily activities (Bergman and Derksen 1977, Eberl and Picman 1993). Areas were estimated by outlining lakes and islet contours (polygons) on a satellite image
(*WorldView 3, color and near-infrared; 0.3m resolution; July 2nd, 2020*) using QGIS software (version 3.16 (QGIS Development Team 2021)). Variables used to describe islet characteristics were not significantly correlated (Spearman correlation coefficients varied between 0.09 and 0.44, all p<0.10; R package *corrplot,* version 0.92 (Wei and Simko, 2021)).

The occurrence of nesting birds on islets was determined annually (summer 2018 and summer 2019). We systematically visited
all known islets in the study area between late June and mid-July, when most birds were incubating. When the islet was occupied by an active nest, we identified the nesting species by direct observation of incubating individuals or with egg/nest characteristics.

## 2.3    Processes generating islets

**Table 1. Brief description of the main geomorphological processes (1 to 5) and biotic processes (6) that can generate islets on Bylot Island (Nunavut, Canada). Criteria used to assign an islet to a specific process are listed in supplementary material (Appendix A).**

| | Process | Description |
|---|---|---|
| 1 | Polygon degradation / Low centered polygon degrading in ridge-like islet | Formed by water isolating raised edge(s) of low centered polygon. Furrow between initial polygons remains while polygons are degrading. |
| 2 | Polygon degradation / Flat centered or High centered polygon degrading in center-like islet | Formed by water isolating polygon center (essentially flat centered polygons in the study area). Center remains. |
| 3 | *Other processes* — Glacial boulders | Large blocks or boulders deposited by glacial drifts, mainly found in postglacial lakes |
| 4 | Raised beach crest degradation | Formed by water isolating degraded raised beach crests (marine deposit aggradation with water recession during coastal water level variation) |
| 5 | Re-exposition or wetland plain degradation (topography, bathymetry) | Formed by water level variation; exposition of uneven surficial deposits following wetland drainage in various lakes and ponds. |
| 6 | Vegetation succession or aggradation | Formed by biotic processes including plant succession or birds, more specifically loons, accumulating vegetation on small shoals to build nests |

We listed all potential abiotic and biotic processes that could generate islets in our study area using the high-resolution *WorldView* satellite image and visual field observations. Based on extensive knowledge and prior research on the surface landforms in the Arctic tundra of Bylot Island (Ellis and Rochefort, 2004; Fortier and Allard, 2004; Fortier et al., 2007), we
listed five main geomorphological processes that can generate islets in our study area (see Table 1). Two main biotic processes





could also lead to islets used by Arctic birds (Table 1) : plant succession may occur in wet plains, and loons are known to accumulate vegetation on small submerged shoals to build nests (Douglas and Reimchen, 1988; Bundy, 1976); both processes can lead to surface aggradation and islet creation. To associate each islet with a specific process, we used a combination of criteria (see the Supplementary Material; Appendix A). Criteria were mainly based on the shape of the water body, the nature

of the surrounding terrain (e.g: *littoral zone, complex polygon wetland, watershed orientation*) and the physical characteristics of the islet.

## 2.4    Statistical analyses

We used a permutation test to assess whether the mean characteristics of the islets selected by the nesting birds were different from those expected at random. For each characteristic (DISTANCE, DEPTH, IsletArea, LakeArea), we compared the mean

value obtained for the islets that were occupied at least once to the distribution of the mean value for 1000 random samples with replacement of all known islets (see details in Appendix B; R package *stats*, version 4.0.3 (R Core Team, 2020).

We used logit-link logistic models with a binomial distribution to evaluate the influence of specific islet characteristics on the probability of occurrence of a bird species on the islet. These models did not incorporate random effects. (R package *lme4*,

version 1.1-27.1; (Bates et al., 2015)). An islet was considered occupied (1) when a nest was found on it at least once during the two-year study period (Manly et al., 2002). Otherwise, it was considered unoccupied (0). All predictor variables were rescaled by their standard deviation. To account for a potential nonlinear effect of DISTANCE and DEPTH, we used distance-weighted functions (Miguet et al., 2017). Following (Carpenter et al., 2010), we first selected the best fitting decay distance function to transform the distance to shore and water depth according to their declining effect (see the full description in

Appendix C).  For each of the three bird species, we used Akaike Information Criterion corrected for small sample size (AICc) to select the best model among a set of models describing the probability of nest occurrence according to islet characteristics (R package *MuMin*, version 1.43.17 (Bartoń, 2020). All models were compared to a null model in which the probability of occurrence had no association with the variables of interest. All candidate models with a ΔAICc lesser than or equal to 2 were considered.


To account for spatial clustering between nest occurrence on islets, we incorporated the geographic coordinates of all islets into our models. The selected models were then tested with and without the coordinates. Parameter estimates were similar with or without spatial variables. Adding or removing LakeArea and IsletArea in the models did not affect our conclusions regarding the effects of DISTANCE and DEPTH (results not shown; but see full model selection in Appendix D). We excluded surface

variables (LakeArea and IsletArea) due to missing data on some islets. This exclusion was done to maximize sample sizes and enhance the accuracy of parameter estimates for the main variables of interest (DISTANCE and DEPTH).



## 3 Results

### 3.1 Islet characteristics and selection

We found 396 islets in the study area. We were able to visit and determine the distance to shore (DISTANCE) and water depth
(DEPTH) for most of them (N = 350 islets used in the subsequent statistical analyzes). Islets were scattered throughout the
entire study area and their characteristics varied substantially (DISTANCE and DEPTH ranging from 1 to 54 m and 3 to >41
cm, respectively; Fig.3b, Appendix E). A total of 84 islets out of 350 (24%) were occupied by a nesting bird (Glaucous gull,
Cackling goose, or Red-throated loon) at least once during the study period. The occupied islets were on average located
further away from the shore and surrounded by deeper water than all available islets in the landscape (DISTANCE: occupied
= 10.6 m ± 7.3 (s.d.), available = 7.4 m ± 6.8 (s.d.); $p_{DISTANCE}$ = 0.002; DEPTH: occupied = 33.6 cm ± 10.6 (s.d.), available =
28.1 cm ± 11.5 (s.d.); $p_{DEPTH}$ = 0.002).

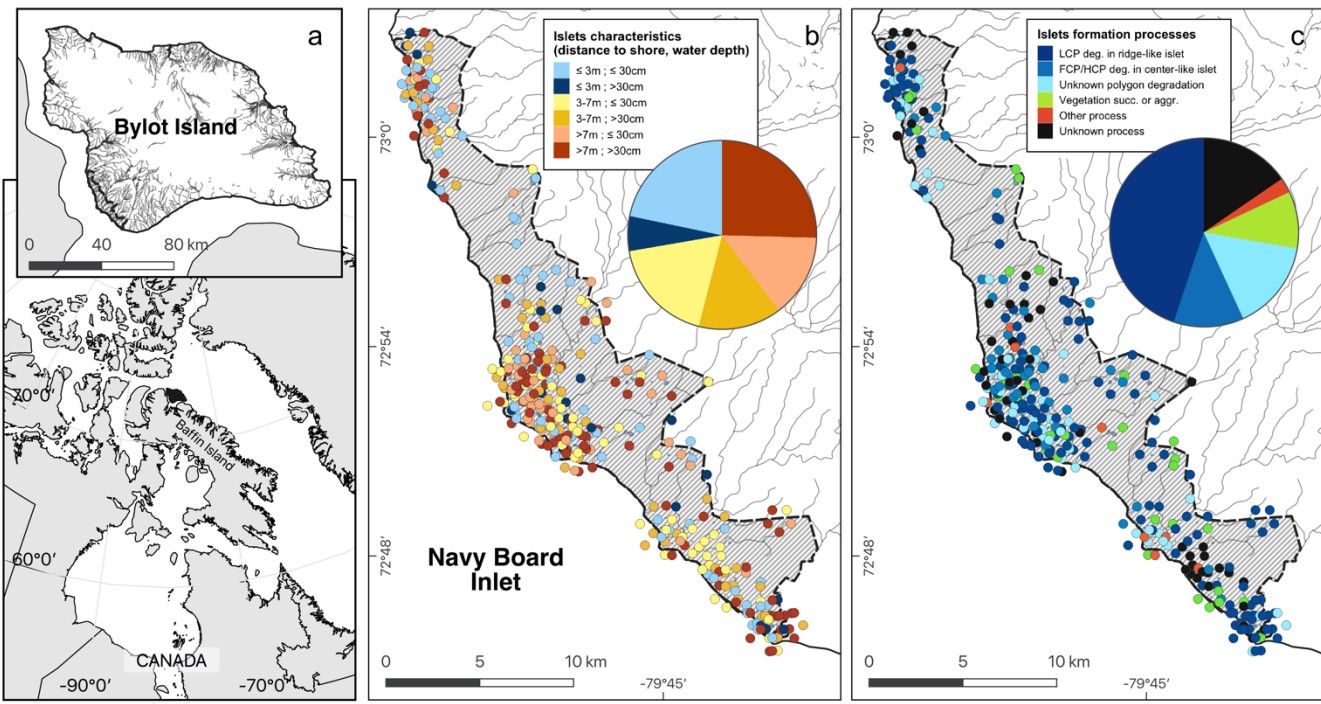

**Figure 3. Maps showing a) the study area (hatched area ~150 km2) on Bylot Island, Nunavut, Canada, b) the spatial distribution of
islets with known characteristics (distance to shore and water depth, N = 350), and c) the geomorphological or biotic processes that
generated these islets based on visual field observations and analysis of a high-resolution satellite image (see also Table 1). The islets
located in dense clusters were jittered in concentric circles around their centroid to reduce overlap. Geomorphological processes:
LCP.deg = Polygon degradation of Low centered polygon degrading in ridge-like islet; FCP/HCP deg. = Polygon degradation of Flat
centered or High centered polygon degrading in center-like islet; Unknown polygon degradation = polygon degradation with
unknown shape; "Other processes" = raised beach crest degradation and wetland plain degradation, or glacial boulders; Unknown
process = unclassified. Biotic processes = Vegetation succession (succ.) or aggradation (aggr.). See table 1 for more detailed
descriptions of the processes.**



The probability of nest occurrence on islets was best explained by the distance to shore and water depth around the islets for all three bird species (for each species, the best fitted model included both DISTANCE and DEPTH; Table 2). All species selected islets less easily accessible to Arctic foxes, with nesting occurrence increasing with DISTANCE and/or DEPTH

(Fig.4). The presence of at least one weighted function in all selected models suggests that the nest occurrence probability for all species increased nonlinearly with distance to shore and/or water depth. For instance, nest occurrence probability increased sharply with distance after the first few meters and gradually stabilized after ~7 meters in gulls (Fig.4a1). Out of 350 islets for which we also had DISTANCE and DEPTH estimates, 315 were visibly discernible in the satellite image. We acquired surface variable estimates (LakeArea and IsletArea) for these islets. Re-running the analyses using this sub-sample did not change our

main results. Whether surface variables were added or not, the effect of DISTANCE and DEPTH remained similar for all species (all candidate models with a ΔAICc ≤ 2 incorporated DISTANCE and/or DEPTH: see full model selection in Appendix D).

**Table 2. Generalized linear model selection of the effects of distance to shore (DISTANCE) and water depth (DEPTH) on bird nest occurrence probability on islets. Left panel report best fitted models and the null model, with model's number of parameters (K), change in AICc from best fitted model (ΔAICc), and Akaike weights (W) for all candidate models with a ΔAICc lesser or equal to 2. Bullseye « ◎ » indicates that a distance weighted function was used for a given variable. The right panel report estimated coefficients of the model with the smallest AICc with their 95%CI. Full model selection is presented in Supplementary material (Appendix D).**

| Model selection | | | | | First model summary | | | |
|---|---|---|---|---|---|---|---|---|
| **Species** | **Model** | **K** | **ΔAICc** | ***W*** | **Parameter** | **Estimate** | **95%CI** | |
| **a) Cackling goose** | DISTANCE◎ + DEPTH | 5 | 0 | 0,27 | Int | 538,5 [ | -313,4 ; | 1427 ] |
| | DEPTH | 4 | 0,1 | 0,26 | Long. | -0,8 [ | -1,8 ; | 0 ] |
| | null | 1 | 14,9 | 0 | Lat. | -0,5 [ | -1,3 ; | 0,3 ] |
| | | | | | DISTANCE* | 4,9 [ | -1,3 ; | 14 ] |
| | | | | | DEPTH | 0,7 [ | **0,2** ; | **1,1** ] |
| **b) Glaucous gull** | DISTANCE◎ + DEPTH◎ | 5 | 0 | 0,44 | Int | -234,9 [ | -1362,5 ; | 925 ] |
| | null | 1 | 38,3 | 0 | Long. | -0,4 [ | -1,6 ; | 0,7 ] |
| | | | | | Lat. | -0,2 [ | -0,9 ; | 1,3 ] |
| | | | | | DISTANCE* | 66,2 [ | **34** ; | **105,9** ] |
| | | | | | DEPTH* | 5,5 [ | **0,2** ; | **12,1** ] |
| **c) Red-throated loon** | DISTANCE + DEPTH◎ | 5 | 0 | 0,29 | Int | -476,4 [ | -1424 ; | 471 ] |
| | DISTANCE | 4 | 0,9 | 0,19 | Long. | 0,3 [ | -0,6 ; | 1,3 ] |
| | null | 1 | 4,1 | 0,04 | Lat. | 0,5 [ | -0,5 ; | 1,4 ] |
| | | | | | DISTANCE | 0,3 [ | **0** ; | **0,6** ] |
| | | | | | DEPTH* | 3,2 [ | -0,4 ; | 7,1 ] |



**Figure 4. Available islets and probability of nest occurrence on islets as a function of distance to shore and water depth (a1; Glaucous gull, a2; Cackling goose, a3; Red-throated loon). The islets used by nesting birds are shown using dark filled circles. The average characteristics (mean distance to shore and mean water depth) of islets assigned to a specific islet formation process are shown in panel b (error bars show 95% confidence intervals). The number of islets associated with each formation process is indicated in the color legend below panel b. Probabilities were derived from selected models (see also Appendix D).**

## 3.2 Processes generating islets

Most of the islets found in the study area (328 out of 396, 83%) were associated with a specific geomorphological or biotic process using visual criteria (see Appendix F). The vast majority (N =281, 71%) of these islets were generated by polygon degradation (see Appendix G), with almost half (N= 177, 45%) associated specifically to low-centered polygon degradation (Fig.3c). The same pattern was observed among the islets with known DISTANCE and DEPTH (72% were generated by polygon degradation; see Appendix E for details and classification of all known islets). In 68 cases, we couldn't attribute a



specific process because some islets weren't clearly visible on satellite images, and field observations lacked the detail needed for a single process assignment.

Islet characteristics (DISTANCE and DEPTH) were not homogenous for islets generated by different processes (Fig.4b). Islets derived from polygonal degradation were on average surrounded by deeper water and farther from the shore than all islets derived from other processes (all grouped together; Wilcoxon signed rank test; DEPTH: polygonal degradation = 28.7 cm ± 11.7 (s.d.), other processes = 24.1 cm ± 11.1 (s.d.), $p_{DEPTH}$ = 0.013, DISTANCE: polygonal degradation = 8.1 m ± 7.5(s.d.), other processes = 5.7 m ± 3.9 (s.d.), $p_{DISTANCE}$ = 0.041, respectively. See Appendix E4 and E5 for two-by-two comparisons
between all processes).

## 4     Discussion

The presence of refuges in the landscape can be critical for species vulnerable to predation (Berryman et al., 2006) and is known to promote species occurrence and coexistence in Arctic vertebrate communities (Duchesne et al., 2021; Clermont et al., 2021; Léandri-Breton and Bêty, 2020). Many Arctic-nesting birds use islets located in patches of wetlands as refuges
(Stickney et al., 2002; Mickelson, 1975; Dahlén and Eriksson, 2002) but little is known about the processes that generate islets with features preferred by birds. In the present study conducted in the Canadian High Arctic, we found that islet characteristics affect nest site selection by three tundra bird species (Glaucous gull, Cackling goose and Red-throated loon). As expected, birds selected islets located farther from the shore and surrounded by deeper water, which are less accessible to the main nest predator (the Arctic fox). A large proportion (71%) of islets in the landscape were generated by ice-wedge polygon degradation,
which also generated islets on average farther from the shore and surrounded by deeper water compared to those generated by other geomorphological or biotic processes. Few attempts have been made to fully integrate geomorphological attributes or processes in birds nest site selection research (Eveillard-Buchoux et al., 2019). To our knowledge, our study is the first conducted in the Arctic that outlines the key role of polygon degradation in the origin of refuges preferred by some prey species.

### 4.1     Physical characteristics and nest site selection

Our results showing the effects of the water depth and the distance to shore on nest site selection are consistent with previous studies conducted on waterfowl (Giroux, 1981; Hammond and Mann, 1956; Lokemoen and Woodward, 1992) and loons (Eberl, 1993) across North America. However, very few studies were conducted on Arctic-nesting birds and at the microhabitat scale like ours (Dahlén and Eriksson, 2002; Weiser and Gilchrist, 2020). Nest site selection can be influenced by several factors
that were not considered in our study. For example, site selection by loons can depend on lake or pond characteristics (e.g. bottom topography, looseness of pond floor, distance to the ocean (Douglas and Reimchen, 1988; Eberl, 1993)). Adding such variables in our analyses would likely improve our ability to explain the probability of nest occurrence on islets. Furthermore,



we could not account for inter-annual variations in water levels for studied water bodies. This variation may affect islet characteristics between years, and therefore their probability of use.

## 4.2    Physical characteristics and predator encounter probability

Fine-scale habitat selection is often related to predator avoidance (3[rd] and 4[th]scale; (Johnson, 1980; Eichholz and Elmberg, 2014)) and our results support the hypothesis that birds select nesting sites according to physical characteristics that reduce the probability of encountering their main nest predator. Tundra species using islets as micro-habitat refuges can partly escape predator-mediated indirect effects generated by changes in the abundance of other prey species, and increase their persistence in a landscape characterised by high predation risk (Duchesne et al., 2021; Clermont et al., 2021). The use of islets or islands as refuges can increase nesting success likely due to a reduced access or a lower abundance of predators (Kellett et al., 2003). The quality of islets should thus be based on physical characteristics that impede predator movements. Several studies show a decrease in the probability of encountering terrestrial mesopredators (such as foxes, skunks, coyotes and badgers) with an increased distance to shore, as well as increased water depth (Lokemoen and Woodward, 1992; Zoellick et al., 2004; Strang, 1976). These physical parameters likely reduce the accessibility of nests on islets because mammalian predators must swim to reach them (Mickelson, 1975).

Arctic foxes, as other mesopredators, are generally reluctant to swim, either in open water (Petersen, 1990) or through water channels (Zoellick et al., 2004). Moreover, walking in mud seems to be a deterrent for Arctic foxes (S.Lai, pers. obs). A complete immersion in cold water or mud followed by a drying or cleaning process (Dickerson et al., 2012) likely generate significant energetic costs in canids. The maximum jumping range and leg length of foxes are likely the two main biomechanical constraints limiting their ability to reach an islet without swimming. For instance, (Strang, 1976) reported that most of the unsuccessful Glaucous gull nests on islets were within fox jumping distance from shore. If the islet is beyond the maximum jumping distance and the water depth exceeds the leg length, the predator should be forced to fully immerse itself to reach the target islet (Zoellick et al., 2004). The non-linear increase in the probability of occurrence of a gull nest after the very first few meters to shore likely reflects these predator biomechanical constraints and potentially outlines a mechanism explaining fine-scale islet selection based on physical characteristics. Enhancing our understanding of Arctic fox movement within wetland areas and the effects of different biomechanical limitations on their ability to access islets would enhance our capacity to assess the quality of refuges within the landscape.

## 4.3    Role of biotic and abiotic processes in generating potential refuges

In the Arctic, cryoturbation and frost cracking are the dominant geomorphological processes that shape the surface of permafrost. These processes lead to pronounced microtopographic reliefs in the form of polygonal networks (Minke et al., 2009; Jorgenson et al., 2015). The degradation of polygons is a cyclical process (French, 2017), resulting in the partial inundation of the landform. Our study highlighted the role of this main geomorphological process, ice-wedge polygon



degradation, in the origin of islets as refuges selected by tundra nesting birds. This is likely a result of the study area's inherent structure, which seems to be representative of wet lowlands throughout the Canadian Arctic. Indeed, the low elevation as well as the predominant arrangement of plateaus, flat lowlands and depressions throughout the southwest plain of Bylot Island have allowed for the formation of multiple polygon complexes, created by the growth of ice wedges, with a significant water supply over time (Gauthier et al., 2013).


Biotic processes such as vegetation aggradation or succession are the second most common processes that generated islets in the study area (about 10% of those that could be classified). We may have slightly underestimated the number of islets associated to this category, as they are generally smaller and perhaps harder to interpret in the field or to classify using satellite images. Some of them may have fallen into the category of islets generated by an unknown process. Since plant succession is

triggered by minor variation in water levels with the presence of colonizing plants surrounding the water body (Magnússon et al., 2020), islets derived from plant succession are less likely to be found in deep water or far from the shore (hence less likely to be selected by birds, see above). Islets derived from vegetation aggradation require a biotic activity, here realized by Red-throated loons. These birds are known for building up their nest by gathering mud and decaying vegetation on a shoal in shallow ponds or on emergent grasses and sedges in wet grassy shallow waters, building up "loon-made islands" (Davis, 1972;

Bundy, 1976). Water depth surrounding the islets formed by such processes thus usually remained relatively shallow.

The low-lying southwest plain of Bylot Island is mainly the result of marine, fluvio-glacial and aeolian sediment deposition over tertiary sedimentary rocks, mostly sandstone and shale (Klassen, 1982; Jackson and Davidson, 1975). Therefore, the occurrence of glacial boulders in glacial drifts is rather uncommon in the landscape, which likely explain why few islets were

generated by the presence of such boulders in our study area. Finally, isostatic uplifting, still ongoing in a part of Bylot Island, generated a succession of narrow coastal ridges from raised beaches, between which shallow wetlands were formed (Woo and Young, 2003). Although their degradation generated few islets in the landscape, the close parallel organization of these coastal raised beaches reduced the likelihood of having large distances between an islet and its nearest shore. Hence, islets generated by such process should be less selected by birds.

**4.4    Climate change and refuges sensitivity**

Considering that ice-wedge polygon degradation can generate a high proportion of islets in the Arctic tundra, climate changes will likely affect the availability or quality of such refuges through alterations in surface hydrology or shifts in permafrost structure (Saulnier-Talbot et al., *submitted*). However, predicting future shifts in islet availability poses a formidable challenge due to the complex interplay among factors affecting ice-rich soil dynamics, coupled with the various scales over which these

changes occur (Bouchard et al., 2020; Francis et al., 2009; French, 2017; Nitzbon et al., 2019). Nonetheless, the ongoing warming trend is accompanied by a rise in extreme seasonal temperature fluctuations and hydrological flux variations, which could potentially exacerbate the degradation of ice wedges and the underlying permafrost in the Arctic tundra (Liljedahl et al.,



2016). This could rapidly lead not only to the degradation of polygonal complexes into shallow thermokarst ponds, but also to a positive feedback amplifying the rate at which these changes occur (Jorgenson et al., 2010; Bouchard et al., 2020).


If we acknowledge that the degradation of ice-wedge polygons by thermokarst in ice-rich soils is a natural and inevitable process at both short and long term (French, 2017), a warming-induced increase in the rate of change could additionally impact the availability of islets, contingent upon the specific extent of degradation currently witnessed in wetland areas. In a polygonal environment at an early degradation state, the increase in soil degradation by thermokarst processes could generate a greater

number of islets by isolating polygonal emerged structures during the formation of thermokarst ponds, which can progressively coalesce together (Hopkins, 1949). The opposite scenario could occur in already well-degraded environments where similar processes could accelerate ground subsidence leading to the destruction of existing islets by the coalescence of ponds into thermokarst lakes (Bouchard et al., 2020). In this situation, the overall number of islets could eventually decrease, which would thus represent a loss of habitat structures heavily used by some bird species. The lowlands, wetlands and complex polygonal

systems generally exhibit various levels of degradation in the landscape, as observed within our study area. Additional research that incorporates this heterogeneity is necessary for a more comprehensive assessment of how warming impacts the fate of ice wedge polygons and the availability of islets in the Arctic landscape.

By linking geomorphological processes and wildlife micro-habitat selection, our study provides fine-grained maps of physical

structures that capture ecologically relevant information and improves our knowledge of geodiversity-biodiversity patterns in the Arctic. Making such bridges between abiotic and biotic realms should ultimately improve our understanding of arctic ecosystem trait diversity and vulnerability to environmental changes (Vernham et al., 2023). The persistence of vulnerable prey can be strongly affected by predation in the arctic tundra (Beardsell et al., 2023), and a change in the availability of refuges could affect community trait diversity. Due to their relatively high body and egg size, birds such as loons, Cackling

geese and gulls are likely easy to detect by predators like foxes (Beardsell et al., 2021). However, they do not have the defensive capabilities of larger tundra nesting species, such as greater snow geese and snowy owls (Duchesne et al., 2021). Their heightened vulnerability to nest predation likely explains why they are mainly found nesting on refuges such as islets (Duchesne et al., 2021). Given its influence on refuge availability through ice-wedge polygon degradation, we can reasonably conclude that global warming is likely to alter predator-prey interactions, species occurrence and distribution in the Arctic

landscape.

## 5    Acknowledgments

Field techniques were approved by University of Quebec at Rimouski Animal Care Committee and field research was approved by the Joint Park Management Committee of Sirmilik National Park of Canada.



This research was made possible by the logistical support provided by the Bylot Island field station of the Center for Northern Studies located in Sirmilik National Park (Parks Canada). We are grateful to the community of Mittimatalik, the Mittimatalik Hunters & Trappers Organization and the staff from Sirmilik National Park for supporting the Bylot Island long-term monitoring program. We are grateful to all the people that supported and participated in the Bylot Island long-term monitoring program. This research was logistically supported by the Polar Continental Shelf Program (PCSP) and PCSP staff in Resolute

and has been financially supported by PCSP, the Natural Sciences and Engineering Research Council of Canada (NSERC), the Weston family Award for Northern Research through the Association of Canadian Universities for Northern Studies (ACUNS), the Northern Scientific Training Program (NSTP), the Fonds de recherche du Québec – Nature and Technologies (FRQNT) and the Fondation de l'UQAR.

We thank Esther Levesque, Samuel Gagnon, Alexis Robitaille and Karine Rioux for their insight on geomorphological and biotic processes generating islets, and Frédéric Bouchard as well as Dominique Berteaux for their welcomed suggestions. We also thank Andréanne Beardsell, Kaïla Bêty-Leclerc, Frédéric Dulude-de Broin, Andra Florea, Louis Moisan, Frédéric Letourneux as well as the goose, fox and lemming teams for their ideas and fieldwork. M.-C. Cadieux and M-J. Rioux provided essential support through coordination of field work campaigns and data management. We thank A. Caron who provided

support with the statistical analyses and finally, we have special thanks to Pierre Legagneux for saving precious data.

## 6    Appendices





## 6.1 Appendix A

**Criteria for islet classification**

**Table A1. List of geomorphological processes generating islets on Bylot Island and visual criteria/characteristics used to assign a given islet to a specific process. Example of islets (orange stars) identified on high-resolution satellite image are also shown.**

| Process | Description | Shape | Feature | Surroundings | Coloration | Picture |
|---|---|---|---|---|---|---|
| Polygon degradation / Low centered polygon degrading in ridge-like islet | Formed by water isolating raised edge(s) of low centered polygon. Furrow between initial polygons remains while polygons are degrading. | Small and narrow to large | Visible furrow line | Aligned or isolated | Heterogenous, green to brown, center remain is usually darker, submerged or not |  |
| Polygon degradation / Flat centered or High centered polygon degrading in center-like islet | Formed by water isolating polygon center (essentially flat centered polygons in the study area). Center remains. | Low surface: perimeter ratio, substantial size | No visible furrow line | Bordered by deep and large furrows | Usually, uniform green or brown |  |
| Polygon degradation/ Undefinable polygon | Lack of conclusive evidence | All sizes and shapes | Not enough clues | Clearly in a polygonal wetland area | All types of coloration |  |
| Vegetation aggradation or succession | Formed by various biotic processes including plant succession or birds accumulating vegetation on small shoals to build nests (Bundy 1976, Douglas and Reimchen 1988). They have been validated by field observations. | Really small, circular |  | Close to shore and in shallow ponds (bottom usually visible) | Dark green hue |  |
| Glacial boulders | Large blocks or boulders deposited by glacial drifts, mainly found in postglacial lakes, as deposition of those boulders by marine drifts is unlikely. | Field observations only |  |  |  |  |
| Raised beach crest degradation | Formed by water isolating degraded raised beach crests (marine deposit aggradation with water recession during coastal water levels variation (Muller and Barr 1966)). | Rectangular, long and narrow | Parallel to other crests | In raised beaches and deltas, aligned with other crests (bottom visible) | All types of coloration |  |
| Re-exposition or wetland plain degradation (topography, bathymetry) | Formed by water level variation; exposition of uneven surficial deposits following wetland drainage in various lakes and ponds. | All sizes and shapes | Smooth edges | No polygonal structures, sandy and /or shallow pond's bottom | All types of coloration |  |
| Unidentifiable |  | Lack of conclusive clues or islet not visible on satellite |  |  |  |  |





## 6.2 Appendix B

**Permutation tests for islet characteristics**

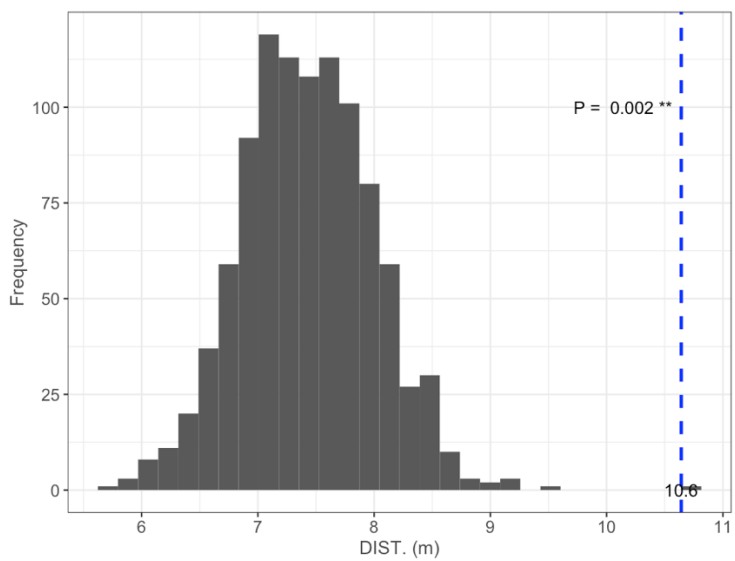

**Figure B1. Permutation test comparing mean DISTANCE of islets (blue broken line) occupied by nesting birds to 1000 random samples (grey bars) of 97 known islets (N = 97, mean = 10.6m; $p_{DISTANCE}$ =0.002\*\*)**

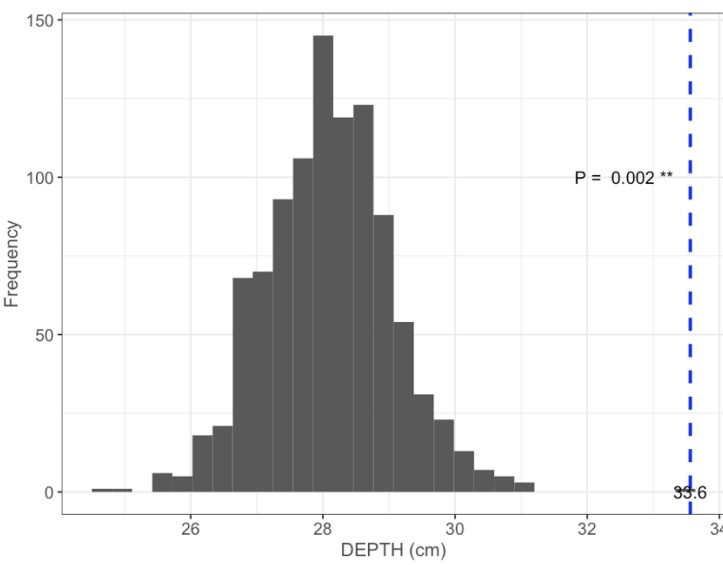

**Figure B2. Permutation test comparing mean DEPTH of islets (blue broken line) occupied by nesting birds (n = 97, mean = 33.6cm) to 1000 random samples (grey bars) of 97 known islets ($p_{DEPTH}$ =0.002\*\*).**



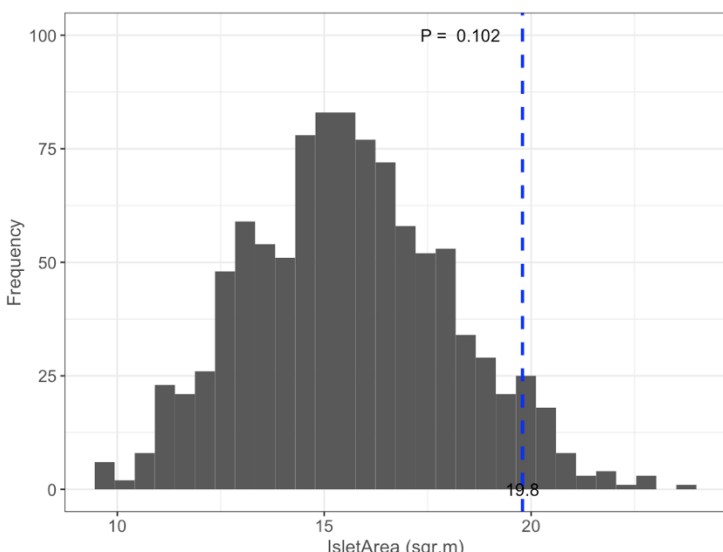

**Figure B3. Permutation test comparing mean IsletArea of islets (blue broken line) occupied by nesting birds (n = 97, mean = 19.8 sqr.m) to 1000 random samples (grey bars) of 97 known islets ($p_{IsletArea}$ =0.102).**

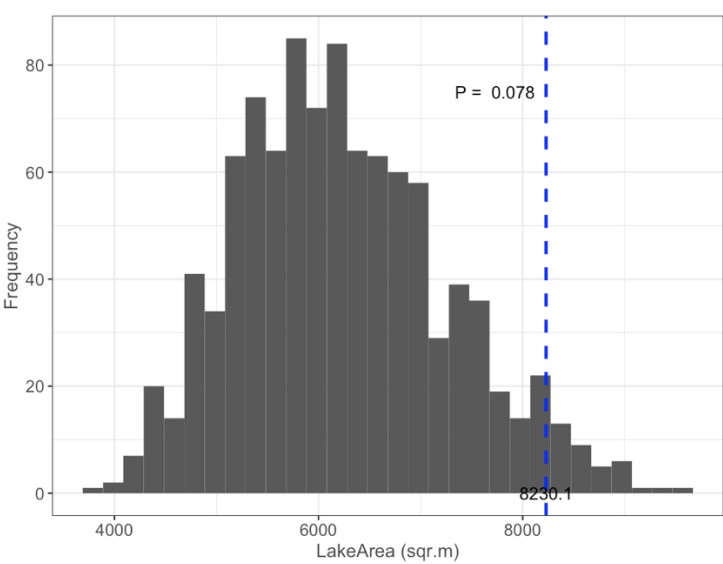

**Figure B4. Permutation test comparing mean LakeArea of islets (blue broken line) occupied by nesting birds (n = 97, mean = 8230.1 sqr.m) to 1000 random samples (grey bars) of 97 known islets ($p_{LakeArea}$ =0.078).**

We were able to outline the islet area (surface) and water body area of 315 islets for which we had DISTANCE and DEPTH. Average IsletArea and LakeArea tended to be greater although not statistically different for occupied and available islets

(IsletArea: P = 0.102; LakeArea: P = 0.078, respectively).



## 6.3 Appendix C

**Statistical analyses**

Distance weighted functions enable the consideration of the continuously declining effect of the surrounding landscape on an ecological response with increasing distance from the point where the response is measured (Miguet et al. 2017). It seemed

adequate to work this way with our variables. We first selected the best fitting decay distance function to transform the distance to shore and water depth according to their declining effect as seen in (Carpenter et al. 2010). We transformed each variable according to the equation $\exp^{-\alpha/\text{DISTANCE or DEPTH}}$, $\alpha$ ranging between the minimum and the maximum rescaled distance or rescaled depth value ( $\alpha_{\text{DISTANCE}}$ min = 0.04, max = 8; $\alpha_{\text{DEPTH}}$ min = 0.25, max = 4). The resulting values ranged from 0 to 1, the highest value representing the effect of the variable at high distance to shore or great water depth.


For each species, we then created a whole set of complete models with geographic coordinates, surface measures and various DISTANCE and DEPTH decay distance functions. Corrected Akaike's Information Criterion (AICc) was used to determine best fitting models. Decay distance functions in complete models presenting the lowest AICc were considered the most competitive and were then used in all competing global models (DISTANCE = $\exp^{-0.04/\text{DISTANCE}}$ for all species and DEPTH

= $\exp^{-1.75/\text{DEPTH}}$ for Cackling goose and DEPTH,  = $\exp^{-4/\text{DEPTH}}$ for Red-throated loon and Glaucous gull; R package MuMIn, (Bartoń 2020)).

For each species, 34 models including surfaces variables were built using every combination of chosen decay distance function with un-transformed islet characteristics. They have been compared to 10 simplified global models excluding surface variables

(see Appendix E). Final models with a AICc ≤ 2 were considered as competitive.





## 6.4 Appendix D

**Full model selections**

**Table D1.1. Generalized linear model selection of the effects of distance to shore (*DISTANCE*) and water depth (*DEPTH*), as well as islet surface and lake surface (*IsletArea* and *LakeArea*, and *Areas* if both surfaces are considered) on Cackling geese nest occurrence probability on islets during year 2018 and 2019 on Bylot Island Southwest plain (N=315). Asterisk « * » indicates that a distance weighted function was used for a given variable. All candidate models are presented with their coefficients estimates, number of parameters (K), change in AICc from best fitted model (ΔAICc), and Akaike weights (W). Models with a ΔAICc ≤ 2 are highlighted in light grey.**

| Models | (Int.) | Long. | Lat. | DISTANCE | DEPTH | DISTANCE* | DEPTH* | IsletArea | LakeArea | Family | K | logLik | AICc | Δ AICc | w |
|---|---|---|---|---|---|---|---|---|---|---|---|---|---|---|---|
| DISTANCE + DEPTH + Areas | 511.59 | -0.83 | -0.47 | 0.43 | 0.67 | | | 0.40 | -1.60 | binomial(logit) | 7 | -106.85 | 228.07 | 0.00 | 0.28 |
| DISTANCE + DEPTH* + Areas | 433.25 | -0.74 | -0.39 | 0.45 | | | 3.89 | 0.41 | -1.62 | binomial(logit) | 7 | -107.53 | 229.43 | 1.35 | 0.14 |
| DISTANCE* + DEPTH + Areas | 490.75 | -0.82 | -0.45 | | 0.69 | 5.85 | | 0.45 | -1.12 | binomial(logit) | 7 | -107.56 | 229.48 | 1.40 | 0.14 |
| DEPTH + Areas | 615.78 | -0.93 | -0.57 | | 0.81 | | | 0.42 | -0.95 | binomial(logit) | 6 | -108.83 | 229.94 | 1.87 | 0.11 |
| DISTANCE + DEPTH + LakeArea | 411.60 | -0.76 | -0.37 | 0.48 | 0.68 | | | | -1.25 | binomial(logit) | 6 | -109.33 | 230.94 | 2.87 | 0.07 |
| DISTANCE* + DEPTH* + Areas | 411.10 | -0.74 | -0.38 | | | 6.10 | 4.02 | 0.45 | -1.11 | binomial(logit) | 7 | -108.38 | 231.13 | 3.05 | 0.06 |
| DEPTH* + Areas | 532.11 | -0.84 | -0.49 | | | | 4.77 | 0.43 | -0.94 | binomial(logit) | 6 | -109.76 | 231.78 | 3.71 | 0.04 |
| DISTANCE + DEPTH* + LakeArea | 331.05 | -0.68 | -0.29 | 0.50 | | | 3.87 | | -1.26 | binomial(logit) | 6 | -110.13 | 232.53 | 4.45 | 0.03 |
| DEPTH+ IsletArea | 522.29 | -0.86 | -0.48 | | 0.76 | | | 0.25 | | binomial(logit) | 5 | -111.74 | 233.68 | 5.61 | 0.02 |
| DISTANCE* + DEPTH + LakeArea | 400.25 | -0.77 | -0.36 | | 0.70 | 5.22 | | | -0.63 | binomial(logit) | 6 | -110.96 | 234.18 | 6.11 | 0.01 |
| DEPTH + LakeArea | 527.02 | -0.88 | -0.48 | | 0.82 | | | | -0.54 | binomial(logit) | 5 | -112.02 | 234.23 | 6.16 | 0.01 |
| DISTANCE* + DEPTH+ IsletArea | 433.46 | -0.78 | -0.39 | | 0.68 | 3.54 | | 0.24 | | binomial(logit) | 6 | -111.20 | 234.67 | 6.60 | 0.01 |
| DEPTH | 485.39 | -0.85 | -0.44 | | 0.78 | | | | | binomial(logit) | 4 | -113.37 | 234.86 | 6.79 | 0.01 |
| DEPTH*+ IsletArea | 447.12 | -0.78 | -0.40 | | | | 4.49 | 0.26 | | binomial(logit) | 5 | -112.62 | 235.43 | 7.35 | 0.01 |
| DISTANCE + DEPTH+ IsletArea | 497.40 | -0.84 | -0.45 | 0.06 | 0.73 | | | 0.24 | | binomial(logit) | 6 | -111.68 | 235.63 | 7.56 | 0.01 |
| DISTANCE* + DEPTH | 383.53 | -0.76 | -0.34 | | 0.69 | 3.88 | | | | binomial(logit) | 5 | -112.72 | 235.64 | 7.57 | 0.01 |
| DISTANCE + Areas | 373.75 | -0.62 | -0.34 | 0.67 | | | | 0.40 | -1.83 | binomial(logit) | 6 | -111.78 | 235.83 | 7.76 | 0.01 |
| DISTANCE + DEPTH | 426.42 | -0.79 | -0.38 | 0.14 | 0.72 | | | | | binomial(logit) | 5 | -112.94 | 236.08 | 8.00 | 0.01 |
| DISTANCE* + DEPTH* + LakeArea | 319.86 | -0.68 | -0.28 | | | 5.49 | 4.02 | | -0.61 | binomial(logit) | 6 | -111.92 | 236.12 | 8.04 | 0.01 |
| DISTANCE* + DEPTH*+ IsletArea | 359.11 | -0.70 | -0.32 | | | 3.78 | 3.94 | 0.25 | | binomial(logit) | 6 | -112.00 | 236.28 | 8.20 | 0.00 |
| DEPTH* + LakeArea | 444.06 | -0.79 | -0.40 | | | | 4.81 | | -0.51 | binomial(logit) | 5 | -113.08 | 236.36 | 8.28 | 0.00 |
| DEPTH* | 409.53 | -0.77 | -0.37 | | | | 4.61 | | | binomial(logit) | 4 | -114.34 | 236.82 | 8.74 | 0.00 |
| DISTANCE + DEPTH*+ IsletArea | 417.14 | -0.75 | -0.37 | 0.07 | | | 4.30 | 0.24 | | binomial(logit) | 6 | -112.51 | 237.29 | 9.21 | 0.00 |
| DISTANCE* + DEPTH* | 307.84 | -0.68 | -0.27 | | | 4.15 | 3.98 | | | binomial(logit) | 5 | -113.62 | 237.43 | 9.35 | 0.00 |
| DISTANCE + DEPTH* | 347.83 | -0.70 | -0.31 | 0.15 | | | 4.20 | | | binomial(logit) | 5 | -113.80 | 237.80 | 9.73 | 0.00 |
| DISTANCE* + Areas | 295.23 | -0.59 | -0.27 | | | 10.18 | | 0.42 | -1.04 | binomial(logit) | 6 | -112.85 | 237.97 | 9.90 | 0.00 |
| DISTANCE+ LakeArea | 284.56 | -0.57 | -0.25 | 0.70 | | | | | -1.46 | binomial(logit) | 5 | -114.35 | 238.89 | 10.82 | 0.00 |
| DISTANCE* + LakeArea | 204.64 | -0.53 | -0.18 | | | 9.99 | | | -0.57 | binomial(logit) | 5 | -116.30 | 242.80 | 14.73 | 0.00 |
| DISTANCE*+ IsletArea | 256.23 | -0.57 | -0.23 | | | 7.83 | | 0.24 | | binomial(logit) | 5 | -116.36 | 242.91 | 14.84 | 0.00 |
| DISTANCE* | 203.24 | -0.54 | -0.17 | | | 8.50 | | | | binomial(logit) | 4 | -118.00 | 244.13 | 16.06 | 0.00 |
| DISTANCE + IsletArea | 350.22 | -0.62 | -0.31 | 0.24 | | | | 0.21 | | binomial(logit) | 5 | -117.97 | 246.13 | 18.06 | 0.00 |
| DISTANCE | 298.97 | -0.59 | -0.26 | 0.30 | | | | | | binomial(logit) | 4 | -119.04 | 246.22 | 18.14 | 0.00 |
| spatial | 419.01 | -0.69 | -0.38 | | | | | | | binomial(logit) | 3 | -121.52 | 249.11 | 21.04 | 0.00 |
| null | -1.87 | | | | | | | | | binomial(logit) | 1 | -123.69 | 249.40 | 21.32 | 0.00 |





**Table D1.2. Generalized linear model selection of the effects of distance to shore (*DISTANCE*) and water depth (*DEPTH*) on Cackling geese nest occurrence probability on islets during year 2018 and 2019 on Bylot Island Southwest plain (N=350). Asterisk « * » indicates that a distance weighted function was used for a given variable. All candidate models are presented with their coefficients estimates, number of parameters (K), change in AICc from best fitted model (ΔAICc), and Akaike weights (W). Models with a ΔAICc ≤ 2 are highlighted in light grey.**

| Models | (Int.) | Long. | Lat. | DISTANCE | DEPTH | DISTANCE* | DEPTH* | Family | K | logLik | AICc | Δ AICc | w |
|---|---|---|---|---|---|---|---|---|---|---|---|---|---|
| DISTANCE* + DEPTH | 538.52 | -0.85 | -0.50 | | 0.66 | 4.85 | | binomial(logit) | 5 | -116.88 | 243.93 | 0.00 | 0.27 |
| DEPTH | 676.11 | -0.97 | -0.63 | | 0.78 | | | binomial(logit) | 4 | -117.96 | 244.04 | 0.11 | 0.26 |
| DISTANCE + DEPTH | 594.48 | -0.89 | -0.55 | 0.17 | 0.71 | | | binomial(logit) | 5 | -117.28 | 244.72 | 0.79 | 0.18 |
| DISTANCE* + DEPTH* | 459.47 | -0.77 | -0.42 | | | 5.20 | 3.79 | binomial(logit) | 5 | -117.77 | 245.71 | 1.77 | 0.11 |
| DEPTH* | 598.08 | -0.88 | -0.55 | | | | 4.61 | binomial(logit) | 4 | -118.99 | 246.10 | 2.16 | 0.09 |
| DISTANCE + DEPTH* | 513.40 | -0.80 | -0.47 | 0.19 | | | 4.09 | binomial(logit) | 5 | -118.15 | 246.47 | 2.54 | 0.08 |
| DISTANCE* | 321.45 | -0.62 | -0.29 | | | 9.28 | | binomial(logit) | 4 | -121.79 | 251.70 | 7.77 | 0.01 |
| DISTANCE | 416.85 | -0.66 | -0.38 | 0.34 | | | | binomial(logit) | 4 | -123.14 | 254.39 | 10.46 | 0.00 |
| spatial | 551.35 | -0.78 | -0.51 | | | | | binomial(logit) | 3 | -126.21 | 258.48 | 14.55 | 0.00 |
| null | -1.99 | | | | | | | binomial(logit) | 1 | -128.42 | 258.86 | 14.92 | 0.00 |

**Table D1.3. Summary of the best fitted model of the effects of distance to shore (*DISTANCE*) and water depth (*DEPTH*), as well as islet surface and lake surface (*IsletArea* and *LakeArea*, and *Areas* if both surfaces are considered) on Cackling geese nest occurrence probability on islets during year 2018 and 2019 on Bylot Island Southwest plain (N=315). Asterisk « * » indicates that a distance weighted function was used for a given variable. We report estimated coefficients of the model with the smallest AICc with their 95%CI.**

| First model summary | | | | | |
|---|---|---|---|---|---|
| **Model** | **Parameter** | **Estimate** | **95%CI** | | |
| DISTANCE + DEPTH + Areas | Int | 511,6 [ | -364 ; | 1427 ] |
| | Long. | -0,8 [ | -1,8 ; | 0,1 ] |
| | Lat. | -0,5 [ | -1,3 ; | 0,4 ] |
| | DISTANCE | 0,4 [ | **0** ; | **0,9** ] |
| | DEPTH | 0,7 [ | **0,2** ; | **1,1** ] |
| | IsletArea | 0,4 [ | **0** ; | **0,8** ] |
| | LakeArea | -1,6 [ | **-3** ; | **-0,5** ] |



**Table D2.1. Generalized linear model selection of the effects of distance to shore (*DISTANCE*) and water depth (*DEPTH*), as well as islet surface and lake surface (*IsletArea* and *LakeArea*, and *Areas* if both surfaces are considered) on Glaucous gulls' nest occurrence probability on islets during year 2018 and 2019 on Bylot Island Southwest plain (N=315). Asterisk « * » indicates that a distance weighted function was used for a given variable. All candidate models are presented with their coefficients estimates, number of parameters (K), change in AICc from best fitted model (ΔAICc), and Akaike weights (W). Models with a ΔAICc ≤ 2 are highlighted in light grey.**

| Models | (Int.) | Long. | Lat. | DISTANCE | DEPTH | DISTANCE* | DEPTH* | IsletArea | LakeArea | Family | K | logLik | AICc | Δ AICc | w |
|---|---|---|---|---|---|---|---|---|---|---|---|---|---|---|---|
| DISTANCE* + DEPTH* + LakeArea | -362.70 | -0.33 | 0.35 | | | 57.05 | 6.57 | | 0.45 | binomial(logit) | 6 | -62.62 | 137.51 | 0.00 | 0.28 |
| DISTANCE* + DEPTH + LakeArea | -354.99 | -0.34 | 0.35 | | 0.77 | 57.30 | | | 0.45 | binomial(logit) | 6 | -62.69 | 137.66 | 0.15 | 0.26 |
| DISTANCE* + DEPTH* + Areas | -391.57 | -0.31 | 0.38 | | | 58.42 | 6.62 | -0.05 | 0.47 | binomial(logit) | 7 | -62.58 | 139.52 | 2.02 | 0.10 |
| DISTANCE* + DEPTH + Areas | -384.54 | -0.32 | 0.37 | | 0.78 | 58.71 | | -0.05 | 0.47 | binomial(logit) | 7 | -62.65 | 139.67 | 2.16 | 0.09 |
| DISTANCE* + DEPTH* | -366.40 | -0.30 | 0.35 | | | 65.75 | 5.86 | | | binomial(logit) | 5 | -65.00 | 140.19 | 2.68 | 0.07 |
| DISTANCE* + DEPTH | -356.12 | -0.31 | 0.34 | | 0.69 | 65.85 | | | | binomial(logit) | 5 | -65.07 | 140.34 | 2.84 | 0.07 |
| DISTANCE* + LakeArea | -510.07 | -0.15 | 0.48 | | | 69.94 | | | 0.38 | binomial(logit) | 5 | -65.45 | 141.09 | 3.58 | 0.05 |
| DISTANCE* + DEPTH*+ IsletArea | -366.76 | -0.30 | 0.35 | | | 65.77 | 5.86 | 0.00 | | binomial(logit) | 6 | -65.00 | 142.27 | 4.76 | 0.03 |
| DISTANCE* + DEPTH+ IsletArea | -356.72 | -0.31 | 0.34 | | 0.69 | 65.88 | | 0.00 | | binomial(logit) | 6 | -65.07 | 142.42 | 4.92 | 0.02 |
| DISTANCE* | -491.69 | -0.14 | 0.45 | | | 76.65 | | | | binomial(logit) | 4 | -67.36 | 142.84 | 5.33 | 0.02 |
| DISTANCE* + Areas | -521.25 | -0.15 | 0.49 | | | 70.62 | | -0.02 | 0.38 | binomial(logit) | 6 | -65.44 | 143.15 | 5.64 | 0.02 |
| DISTANCE*+ IsletArea | -484.35 | -0.14 | 0.45 | | | 76.11 | | 0.02 | | binomial(logit) | 5 | -67.35 | 144.90 | 7.39 | 0.01 |
| DISTANCE + DEPTH* + LakeArea | 34.16 | -0.56 | 0.01 | 0.29 | | | 9.42 | | 0.39 | binomial(logit) | 6 | -70.52 | 153.31 | 15.80 | 0.00 |
| DISTANCE + DEPTH + LakeArea | 42.71 | -0.56 | 0.00 | 0.30 | 1.11 | | | | 0.38 | binomial(logit) | 6 | -70.64 | 153.55 | 16.04 | 0.00 |
| DEPTH* + LakeArea | 210.73 | -0.73 | -0.16 | | | | 11.00 | | 0.47 | binomial(logit) | 5 | -72.16 | 154.52 | 17.01 | 0.00 |
| DEPTH + LakeArea | 221.36 | -0.74 | -0.17 | | 1.29 | | | | 0.46 | binomial(logit) | 5 | -72.33 | 154.85 | 17.35 | 0.00 |
| DISTANCE + DEPTH* + Areas | 34.45 | -0.56 | 0.01 | 0.29 | | | 9.42 | 0.00 | 0.39 | binomial(logit) | 7 | -70.52 | 155.40 | 17.89 | 0.00 |
| DISTANCE + DEPTH + Areas | 41.45 | -0.56 | 0.00 | 0.30 | 1.11 | | | 0.00 | 0.38 | binomial(logit) | 7 | -70.64 | 155.64 | 18.14 | 0.00 |
| DEPTH* + Areas | 233.32 | -0.74 | -0.18 | | | | 10.81 | 0.11 | 0.45 | binomial(logit) | 6 | -71.94 | 156.16 | 18.65 | 0.00 |
| DISTANCE + DEPTH* | 18.54 | -0.51 | 0.02 | 0.41 | | | 8.21 | | | binomial(logit) | 5 | -72.99 | 156.16 | 18.66 | 0.00 |
| DISTANCE + DEPTH | 29.77 | -0.52 | 0.01 | 0.41 | 0.97 | | | | | binomial(logit) | 5 | -73.06 | 156.30 | 18.80 | 0.00 |
| DEPTH + Areas | 243.05 | -0.75 | -0.19 | | 1.27 | | | 0.11 | 0.44 | binomial(logit) | 6 | -72.12 | 156.51 | 19.01 | 0.00 |
| DISTANCE + DEPTH*+ IsletArea | 28.34 | -0.51 | 0.01 | 0.40 | | | 8.20 | 0.02 | | binomial(logit) | 6 | -72.98 | 158.23 | 20.73 | 0.00 |
| DISTANCE + DEPTH+ IsletArea | 38.55 | -0.52 | 0.00 | 0.40 | 0.97 | | | 0.02 | | binomial(logit) | 6 | -73.05 | 158.37 | 20.87 | 0.00 |
| DEPTH* | 244.64 | -0.72 | -0.19 | | | | 9.80 | | | binomial(logit) | 4 | -76.33 | 160.79 | 23.28 | 0.00 |
| DEPTH | 257.46 | -0.73 | -0.21 | | 1.16 | | | | | binomial(logit) | 4 | -76.42 | 160.96 | 23.45 | 0.00 |
| DEPTH*+ IsletArea | 282.03 | -0.74 | -0.23 | | | | 9.56 | 0.19 | | binomial(logit) | 5 | -75.71 | 161.62 | 24.12 | 0.00 |
| DEPTH+ IsletArea | 293.73 | -0.75 | -0.25 | | 1.13 | | | 0.19 | | binomial(logit) | 5 | -75.81 | 161.82 | 24.31 | 0.00 |
| DISTANCE+ LakeArea | -160.89 | -0.27 | 0.19 | 0.51 | | | | | 0.25 | binomial(logit) | 5 | -77.11 | 164.41 | 26.91 | 0.00 |
| DISTANCE | -146.34 | -0.26 | 0.18 | 0.57 | | | | | | binomial(logit) | 4 | -78.43 | 164.98 | 27.47 | 0.00 |
| DISTANCE + Areas | -159.70 | -0.27 | 0.19 | 0.51 | | | | 0.00 | 0.25 | binomial(logit) | 6 | -77.11 | 166.49 | 28.98 | 0.00 |
| DISTANCE + IsletArea | -136.42 | -0.27 | 0.17 | 0.56 | | | | 0.03 | | binomial(logit) | 5 | -78.41 | 167.02 | 29.52 | 0.00 |
| null | -2.45 | | | | | | | | | binomial(logit) | 1 | -87.32 | 176.66 | 39.15 | 0.00 |
| spatial | 148.22 | -0.51 | -0.11 | | | | | | | binomial(logit) | 3 | -85.67 | 177.42 | 39.92 | 0.00 |





**Table D2.2.** Generalized linear model selection of the effects of distance to shore (*DISTANCE*) and water depth (*DEPTH*) on Glaucous gulls' nest occurrence probability on islets during year 2018 and 2019 on Bylot Island Southwest plain (N=350). Asterisk « * » indicates that a distance weighted function was used for a given variable. All candidate models are presented with their coefficients estimates, number of parameters (K), change in AICc from best fitted model (ΔAICc), and Akaike weights (W). Models with a ΔAICc ≤ 2 are highlighted in light grey.

| Models | (Int.) | Long. | Lat. | DISTANCE | DEPTH | DISTANCE* | DEPTH* | Family | K | logLik | AICc | Δ AICc | w |
|---|---|---|---|---|---|---|---|---|---|---|---|---|---|
| DISTANCE* + DEPTH* | -234.87 | -0.38 | 0.21 | | | 66.22 | 5.52 | binomial(logit) | 5 | -66.83 | 143.84 | 0.00 | 0.44 |
| DISTANCE* + DEPTH | -226.07 | -0.38 | 0.20 | | 0.65 | 66.34 | | binomial(logit) | 5 | -66.90 | 143.98 | 0.14 | 0.41 |
| DISTANCE* | -394.37 | -0.20 | 0.36 | | | 76.65 | | binomial(logit) | 4 | -68.92 | 145.95 | 2.11 | 0.15 |
| DISTANCE + DEPTH* | 198.12 | -0.62 | -0.16 | 0.43 | | | 8.05 | binomial(logit) | 5 | -75.43 | 161.04 | 17.20 | 0.00 |
| DISTANCE + DEPTH | 207.63 | -0.63 | -0.17 | 0.43 | 0.95 | | | binomial(logit) | 5 | -75.50 | 161.17 | 17.33 | 0.00 |
| DEPTH* | 459.90 | -0.85 | -0.41 | | | | 9.79 | binomial(logit) | 4 | -79.26 | 166.63 | 22.79 | 0.00 |
| DEPTH | 471.02 | -0.87 | -0.42 | | 1.16 | | | binomial(logit) | 4 | -79.33 | 166.79 | 22.94 | 0.00 |
| DISTANCE | -32.45 | -0.34 | 0.06 | 0.60 | | | | binomial(logit) | 4 | -80.62 | 169.35 | 25.51 | 0.00 |
| null | -2.56 | | | | | | | binomial(logit) | 1 | -90.06 | 182.13 | 38.29 | 0.00 |
| spatial | 278.64 | -0.60 | -0.24 | | | | | binomial(logit) | 3 | -88.58 | 183.22 | 39.38 | 0.00 |

**Table D2.3.** Summary of the best fitted model of the effects of distance to shore (*DISTANCE*) and water depth (*DEPTH*), as well as islet surface and lake surface (*IsletArea* and *LakeArea*, and *Areas* if both surfaces are considered) on Glaucous gulls nest occurrence probability on islets during year 2018 and 2019 on Bylot Island Southwest plain (N=315). Asterisk « * »indicates that a distance weighted function was used for a given variable. We report estimated coefficients of the model with the smallest AICc with their 95%CI.

| First model summary | | | | |
|---|---|---|---|---|
| **Model** | **Parameter** | **Estimate** | **95%CI** | |
| DISTANCE* + DEPTH* + LakeArea | Int | -362,7 [ | -1541 ; | 841 ] |
| | Long. | -0,3 [ | -1,6 ; | 0,8 ] |
| | Lat. | 0,3 [ | -0,8 ; | 1,5 ] |
| | DISTANCE* | 57,1 [ | **26,8** ; | **94,7** ] |
| | DEPTH* | 6,6 [ | **1,1** ; | **13,4** ] |
| | LakeArea | 0,5 [ | **0,1** ; | **0,8** ] |



**Table D3.1. Generalized linear model selection of the effects of distance to shore (*DISTANCE*) and water depth (*DEPTH*), as well as islet surface and lake surface (*IsletArea* and *LakeArea*, and *Areas* if both surfaces are considered) on Red-throated loons nest occurrence probability on islets during year 2018 and 2019 on Bylot Island Southwest plain (N=315). Asterisk « * » indicates that a distance weighted function was used for a given variable. All candidate models are presented with their coefficients estimates, number of parameters (K), change in AICc from best fitted model (ΔAICc), and Akaike weights (W). Models with a ΔAICc ≤ 2 are highlighted in light grey.**

| Models | (Int.) | Long. | Lat. | DISTANCE | DEPTH | DISTANCE* | DEPTH* | IsletArea | LakeArea | Family | K | logLik | AICc | Δ AICc | w |
|---|---|---|---|---|---|---|---|---|---|---|---|---|---|---|---|
| DISTANCE* + DEPTH* | -566.88 | 0.37 | 0.52 | | | 38.51 | 3.85 | | | binomial(logit) | 5 | -74.43 | 159.05 | 0.00 | 0.19 |
| DISTANCE* + DEPTH | -560.15 | 0.37 | 0.52 | | 0.45 | 38.60 | | | | binomial(logit) | 5 | -74.49 | 159.17 | 0.13 | 0.18 |
| DISTANCE* | -622.84 | 0.47 | 0.57 | | | 45.34 | | | | binomial(logit) | 4 | -75.73 | 159.59 | 0.54 | 0.14 |
| DISTANCE* + DEPTH* + LakeArea | -568.90 | 0.38 | 0.53 | | | 36.61 | 3.93 | | 0.12 | binomial(logit) | 6 | -74.29 | 160.84 | 1.79 | 0.08 |
| DISTANCE* + DEPTH + LakeArea | -562.35 | 0.37 | 0.52 | | 0.46 | 36.70 | | | 0.12 | binomial(logit) | 6 | -74.35 | 160.97 | 1.92 | 0.07 |
| DISTANCE* + DEPTH*+ IsletArea | -591.62 | 0.38 | 0.55 | | | 39.77 | 3.89 | -0.07 | | binomial(logit) | 6 | -74.36 | 160.98 | 1.94 | 0.07 |
| DISTANCE* + DEPTH+ IsletArea | -584.60 | 0.38 | 0.54 | | 0.45 | 39.87 | | -0.07 | | binomial(logit) | 6 | -74.42 | 161.11 | 2.06 | 0.07 |
| DISTANCE* + LakeArea | -625.97 | 0.47 | 0.57 | | | 43.99 | | | 0.09 | binomial(logit) | 5 | -75.64 | 161.47 | 2.42 | 0.06 |
| DISTANCE*+ IsletArea | -640.39 | 0.47 | 0.58 | | | 46.38 | | -0.06 | | binomial(logit) | 5 | -75.68 | 161.55 | 2.50 | 0.05 |
| DISTANCE* + DEPTH* + Areas | -597.70 | 0.39 | 0.56 | | | 37.87 | 3.98 | -0.08 | 0.14 | binomial(logit) | 7 | -74.18 | 162.73 | 3.68 | 0.03 |
| DISTANCE* + DEPTH + Areas | -590.91 | 0.38 | 0.55 | | 0.47 | 37.96 | | -0.09 | 0.14 | binomial(logit) | 7 | -74.24 | 162.85 | 3.80 | 0.03 |
| DISTANCE* + Areas | -646.58 | 0.48 | 0.59 | | | 45.05 | | -0.07 | 0.10 | binomial(logit) | 6 | -75.57 | 163.41 | 4.36 | 0.02 |
| DISTANCE + DEPTH* | -361.75 | 0.25 | 0.36 | 0.35 | | | 5.66 | | | binomial(logit) | 5 | -79.16 | 168.52 | 9.47 | 0.00 |
| DISTANCE + DEPTH | -353.29 | 0.24 | 0.35 | 0.35 | 0.67 | | | | | binomial(logit) | 5 | -79.22 | 168.64 | 9.59 | 0.00 |
| DISTANCE + DEPTH*+ IsletArea | -406.89 | 0.28 | 0.40 | 0.40 | | | 5.69 | -0.14 | | binomial(logit) | 6 | -78.93 | 170.13 | 11.09 | 0.00 |
| DISTANCE + DEPTH+ IsletArea | -398.44 | 0.27 | 0.39 | 0.40 | 0.67 | | | -0.14 | | binomial(logit) | 6 | -78.99 | 170.25 | 11.20 | 0.00 |
| DISTANCE + DEPTH* + LakeArea | -360.13 | 0.25 | 0.35 | 0.31 | | | 5.81 | | 0.12 | binomial(logit) | 6 | -79.02 | 170.31 | 11.26 | 0.00 |
| DISTANCE + DEPTH + LakeArea | -352.02 | 0.24 | 0.35 | 0.31 | 0.68 | | | | 0.12 | binomial(logit) | 6 | -79.09 | 170.44 | 11.39 | 0.00 |
| DEPTH* | -183.54 | 0.07 | 0.18 | | | | 7.05 | | | binomial(logit) | 4 | -81.51 | 171.15 | 12.10 | 0.00 |
| DEPTH | -173.43 | 0.06 | 0.17 | | 0.83 | | | | | binomial(logit) | 4 | -81.59 | 171.30 | 12.25 | 0.00 |
| DEPTH* + LakeArea | -202.93 | 0.08 | 0.20 | | | | 7.22 | | 0.23 | binomial(logit) | 5 | -80.76 | 171.70 | 12.66 | 0.00 |
| DEPTH + LakeArea | -193.45 | 0.07 | 0.19 | | 0.85 | | | | 0.22 | binomial(logit) | 5 | -80.84 | 171.88 | 12.83 | 0.00 |
| DISTANCE + DEPTH* + Areas | -404.83 | 0.27 | 0.40 | 0.36 | | | 5.83 | -0.14 | 0.13 | binomial(logit) | 7 | -78.78 | 171.92 | 12.87 | 0.00 |
| DISTANCE + DEPTH + Areas | -396.78 | 0.26 | 0.39 | 0.36 | 0.69 | | | -0.15 | 0.13 | binomial(logit) | 7 | -78.84 | 172.04 | 12.99 | 0.00 |
| DISTANCE | -436.92 | 0.40 | 0.42 | 0.48 | | | | | | binomial(logit) | 4 | -82.28 | 172.69 | 13.64 | 0.00 |
| DEPTH*+ IsletArea | -179.31 | 0.07 | 0.18 | | | | 7.00 | 0.04 | | binomial(logit) | 5 | -81.49 | 173.17 | 14.12 | 0.00 |
| DEPTH+ IsletArea | -169.39 | 0.06 | 0.17 | | 0.83 | | | 0.04 | | binomial(logit) | 5 | -81.57 | 173.32 | 14.28 | 0.00 |
| DEPTH* + Areas | -202.49 | 0.08 | 0.20 | | | | 7.22 | 0.00 | 0.23 | binomial(logit) | 6 | -80.75 | 173.78 | 14.73 | 0.00 |
| DEPTH + Areas | -193.10 | 0.07 | 0.19 | | 0.85 | | | 0.00 | 0.22 | binomial(logit) | 6 | -80.84 | 173.96 | 14.91 | 0.00 |
| DISTANCE + IsletArea | -471.04 | 0.42 | 0.46 | 0.53 | | | | -0.14 | | binomial(logit) | 5 | -82.06 | 174.31 | 15.26 | 0.00 |
| DISTANCE+ LakeArea | -437.62 | 0.41 | 0.42 | 0.47 | | | | | 0.05 | binomial(logit) | 5 | -82.26 | 174.71 | 15.66 | 0.00 |
| DISTANCE + Areas | -471.79 | 0.42 | 0.46 | 0.51 | | | | -0.14 | 0.06 | binomial(logit) | 6 | -82.02 | 176.32 | 17.27 | 0.00 |
| null | -2.45 | | | | | | | | | binomial(logit) | 1 | -87.32 | 176.66 | 17.61 | 0.00 |
| spatial | -205.41 | 0.19 | 0.20 | | | | | | | binomial(logit) | 3 | -87.25 | 180.57 | 21.52 | 0.00 |







**Table D3.2. Generalized linear model selection of the effects of distance to shore (*DISTANCE*) and water depth (*DEPTH*) on Red-throated loons nest occurrence probability on islets during year 2018 and 2019 on Bylot Island Southwest plain (N=350). Asterisk « * » indicates that a distance weighted function was used for a given variable. All candidate models are presented with their coefficients estimates, number of parameters (K), change in AICc from best fitted model (ΔAICc), and Akaike weights (W). Models with a ΔAICc ≤ 2 are highlighted in light grey.**

| Models | (Int.) | Long. | Lat. | DISTANCE | DEPTH | DISTANCE* | DEPTH* | Family | K | logLik | AICc | Δ AICc | w |
|---|---|---|---|---|---|---|---|---|---|---|---|---|---|
| DISTANCE + DEPTH* | -476.36 | 0.34 | 0.47 | 0.34 | | | 3.17 | binomial(logit) | 5 | -100.91 | 211.99 | 0.00 | 0.29 |
| DISTANCE + DEPTH | -475.97 | 0.34 | 0.47 | 0.34 | 0.36 | | | binomial(logit) | 5 | -101.03 | 212.23 | 0.25 | 0.26 |
| DISTANCE | -528.68 | 0.43 | 0.52 | 0.43 | | | | binomial(logit) | 4 | -102.39 | 212.89 | 0.91 | 0.19 |
| DEPTH* | -297.98 | 0.18 | 0.30 | | | | 4.45 | binomial(logit) | 4 | -103.31 | 214.73 | 2.75 | 0.07 |
| DEPTH | -295.49 | 0.17 | 0.29 | | 0.52 | | | binomial(logit) | 4 | -103.48 | 215.08 | 3.09 | 0.06 |
| DISTANCE* + DEPTH* | -365.00 | 0.23 | 0.36 | | | 2.63 | 3.84 | binomial(logit) | 5 | -102.93 | 216.04 | 4.05 | 0.04 |
| null | -2.30 | | | | | | | binomial(logit) | 1 | -107.04 | 216.09 | 4.11 | 0.04 |
| DISTANCE* + DEPTH | -363.61 | 0.23 | 0.36 | | 0.44 | 2.68 | | binomial(logit) | 5 | -103.08 | 216.34 | 4.36 | 0.03 |
| DISTANCE* | -445.53 | 0.32 | 0.43 | | | 5.15 | | binomial(logit) | 4 | -105.12 | 218.35 | 6.37 | 0.01 |
| spatial | -316.27 | 0.24 | 0.31 | | | | | binomial(logit) | 3 | -106.76 | 219.60 | 7.61 | 0.01 |

**Table D3.3. Summary of the best fitted model of the effects of distance to shore (*DISTANCE*) and water depth (*DEPTH*), as well as islet surface and lake surface (*IsletArea* and *LakeArea*, and *Areas* if both surfaces are considered) on Red-throated loons nest occurrence probability on islets during year 2018 and 2019 on Bylot Island Southwest plain (N=315). Asterisk « * » indicates that a distance weighted function was used for a given variable. We report estimated coefficients of the model with the smallest AICc with their 95%CI.**

| First model summary | | | |
|---|---|---|---|
| **Model** | **Parameter** | **Estimate** | **95%CI** |
| DISTANCE* + DEPTH* | Int | -566,9 [ | -1635 ; 530,9 ] |
| | Long. | 0,4 [ | -0,7 ; 1,4 ] |
| | Lat. | 0,5 [ | -0,6 ; 1,6 ] |
| | DISTANCE* | 38,5 [ | **14,4 ; 69,4** ] |
| | DEPTH* | 3,8 [ | -0,8 ; 9,3 ] |



## 6.5    Appendix E

**Variation of islet characteristics**

**Table E1. All known (N= 396) islet characteristics and proportion of islets occupied by a nesting bird (N.occ; % occ.) at least once over two years for each islet formation process.**

| Process | Processes | | Occupation | |
|---|---|---|---|---|
| | N | % | N.occ | % occ |
| LCP deg. in ridge-like islet | 177 | 45 | 45 | 11 |
| FCP/HCP deg. in center-like islet | 47 | 12 | 8 | 2 |
| Unknown polygon degradation | 57 | 14 | 16 | 4 |
| Vegetation succ. or aggr. | 38 | 10 | 11 | 3 |
| Other process | 9 | 2 | 3 | 1 |
| Unknown process | 68 | 17 | 14 | 4 |


**Table E2. Characteristics for all islets with known DISTANCE (in meters) and DEPTH (in centimeters; N = 350) with proportion of islets occupied by a nesting bird (N.occ; % occ.) at least once over two years for each islet formation process. Mean ± SD and range (min – max) are provided for each characteristic.**

| Process | Processes | | DISTANCE | | | | DEPTH | | | | Occupation | |
|---|---|---|---|---|---|---|---|---|---|---|---|---|
| | N | % | meanD | sdD | minD | maxD | meanP | sdP | minP | maxP | N.occ | % occ |
| LCP deg. in ridge-like islet | 157 | 45 | 7 | 6 | 0 | 35 | 28 | 12 | 5 | 41 | 38 | 11 |
| FCP/HCP deg. in center-like islet | 42 | 12 | 12 | 13 | 0 | 54 | 25 | 13 | 4 | 41 | 8 | 2 |
| Unknown polygon degradation | 54 | 15 | 8 | 5 | 1 | 25 | 33 | 9 | 10 | 41 | 15 | 4 |
| Vegetation succ. or aggr. | 34 | 10 | 6 | 4 | 1 | 18 | 24 | 11 | 5 | 41 | 10 | 3 |
| Other process | 9 | 3 | 5 | 3 | 1 | 10 | 25 | 11 | 7 | 41 | 3 | 1 |
| Unknown process | 54 | 15 | 5 | 4 | 0 | 20 | 28 | 11 | 3 | 41 | 10 | 3 |


**Table E3. Characteristics for all islets with known DISTANCE (in meters), DEPTH (in centimeters), IsletArea (in square meters) and LakeArea (in square meters; N = 315) with proportion of islets occupied by a nesting bird (N.occ; % occ.) at least once over two years for each islet formation process. Mean ± SD and range (min – max) are provided for each characteristic.**

| Process | Processes | | DISTANCE | | | | DEPTH | | | | IsletArea | | | | LakeArea | | | | Occupation | |
|---|---|---|---|---|---|---|---|---|---|---|---|---|---|---|---|---|---|---|---|---|
| | N | % | meanD | sdD | minD | maxD | meanP | sdP | minP | maxP | meanI | sdI | minI | maxI | meanL | sdL | minL | maxL | N.occ | % occ |
| LCP deg. in ridge-like islet | 155 | 49 | 7 | 6 | 0 | 35 | 28 | 12 | 5 | 41 | 17 | 29 | 1 | 200 | 4738 | 4804 | 321 | 25424 | 38 | 12 |
| FCP/HCP deg. in center-like islet | 42 | 13 | 12 | 13 | 0 | 54 | 25 | 13 | 4 | 41 | 34 | 46 | 2 | 188 | 10471 | 10178 | 709 | 25424 | 8 | 3 |
| Unknown polygon degradation | 53 | 17 | 8 | 5 | 1 | 25 | 34 | 9 | 10 | 41 | 10 | 11 | 1 | 70 | 3764 | 3115 | 556 | 16290 | 15 | 5 |
| Vegetation succ. or aggr. | 33 | 10 | 6 | 4 | 1 | 18 | 24 | 11 | 7 | 41 | 5 | 7 | 0 | 29 | 8550 | 19084 | 193 | 78711 | 9 | 3 |
| Other process | 8 | 3 | 5 | 3 | 1 | 10 | 25 | 12 | 7 | 41 | 15 | 15 | 1 | 47 | 28765 | 52708 | 737 | 140886 | 2 | 1 |
| Unknown process | 24 | 8 | 6 | 4 | 1 | 15 | 29 | 12 | 3 | 41 | 5 | 7 | 1 | 29 | 3005 | 4926 | 844 | 25424 | 5 | 2 |






**Table E4. Wilcoxon signed rank test comparing mean distances to shore between categories of processes generating islets (N=350). Sample size (NP1, NP2), Wilcoxon signed rank test statistic ($W_i$), p-values as well as their significance are shown.**

| Category 1 | Category 2 | NP1 | NP2 | $W_i$ | p | p.signif |
|---|---|---|---|---|---|---|
| LCP deg. in ridge-like islet | FCP/HCP deg. in center-like islet | 157 | 42 | 2719.5 | 0.081 | ns |
| LCP deg. in ridge-like islet | Unknown polygon degradation | 157 | 54 | 3462.0 | 0.044 | * |
| LCP deg. in ridge-like islet | Vegetation succ. or aggr. | 157 | 34 | 2952.0 | 0.332 | ns |
| LCP deg. in ridge-like islet | Other process | 157 | 9 | 809.5 | 0.463 | ns |
| LCP deg. in ridge-like islet | Unknown process | 157 | 54 | 4999.0 | 0.049 | * |
| FCP/HCP deg. in center-like islet | Unknown polygon degradation | 42 | 54 | 1154.5 | 0.882 | ns |
| FCP/HCP deg. in center-like islet | Vegetation succ. or aggr. | 42 | 34 | 916.0 | 0.035 | * |
| FCP/HCP deg. in center-like islet | Other process | 42 | 9 | 246.0 | 0.161 | ns |
| FCP/HCP deg. in center-like islet | Unknown process | 42 | 54 | 1529.5 | 0.003 | ** |
| Unknown polygon degradation | Vegetation succ. or aggr. | 54 | 34 | 1232.5 | 0.007 | ** |
| Unknown polygon degradation | Other process | 54 | 9 | 341.5 | 0.053 | ns |
| Unknown polygon degradation | Unknown process | 54 | 54 | 2048.0 | 0.000 | *** |
| Vegetation succ. or aggr. | Other process | 34 | 9 | 152.0 | 0.988 | ns |
| Vegetation succ. or aggr. | Unknown process | 34 | 54 | 986.0 | 0.561 | ns |
| Other process | Unknown process | 9 | 54 | 269.0 | 0.615 | ns |


**Table E5. Wilcoxon signed rank test comparing mean water depths between categories of processes generating islets (N=350). Sample size (NP1, NP2), Wilcoxon signed rank test statistic ($W_i$), p-values as well as their significance are shown.**

| Category 1 | Category 2 | NP1 | NP1 | $W_i$ | p | p.signif |
|---|---|---|---|---|---|---|
| LCP deg. in ridge-like islet | FCP/HCP deg. in center-like islet | 157 | 42 | 3693.5 | 0.224 | ns |
| LCP deg. in ridge-like islet | Unknown polygon degradation | 157 | 54 | 3358.5 | 0.020 | * |
| LCP deg. in ridge-like islet | Vegetation succ. or aggr. | 157 | 34 | 3226.0 | 0.053 | ns |
| LCP deg. in ridge-like islet | Other process | 157 | 9 | 834.0 | 0.357 | ns |
| LCP deg. in ridge-like islet | Unknown process | 157 | 54 | 4228.0 | 0.978 | ns |
| FCP/HCP deg. in center-like islet | Unknown polygon degradation | 42 | 54 | 764.0 | 0.005 | ** |
| FCP/HCP deg. in center-like islet | Vegetation succ. or aggr. | 42 | 34 | 761.5 | 0.621 | ns |
| FCP/HCP deg. in center-like islet | Other process | 42 | 9 | 193.0 | 0.930 | ns |
| FCP/HCP deg. in center-like islet | Unknown process | 42 | 54 | 978.0 | 0.245 | ns |
| Unknown polygon degradation | Vegetation succ. or aggr. | 54 | 34 | 1343.0 | 0.0002 | *** |
| Unknown polygon degradation | Other process | 54 | 9 | 355.5 | 0.025 | * |
| Unknown polygon degradation | Unknown process | 54 | 54 | 1801.0 | 0.032 | * |
| Vegetation succ. or aggr. | Other process | 34 | 9 | 148.0 | 0.893 | ns |
| Vegetation succ. or aggr. | Unknown process | 34 | 54 | 702.0 | 0.062 | ns |
| Other process | Unknown process | 9 | 54 | 190.0 | 0.296 | ns |



## 6.6    Appendix F

**Processes for all known islets**

**Figure F1. Map showing a) the study area (hatched area ~150 km²) on Bylot Island, Nunavut, Canada and b) the geomorphological or biotic processes that generated these islets (N=396) based on visual field observations and analysis of a high-resolution satellite image and field observations (see also Table 1). the islets located in dense clusters were jittered in concentric circles around their**
**centroid to reduce overlap. Geomorphological processes: *LCP.deg* = Polygon degradation of Low centered polygon degrading in ridge-like islet; *FCP/HCP deg.* = Polygon degradation of Flat centered or High centered polygon degrading in center-like islet; *Unknown polygon degradation* = polygon degradation with unknown shape; "Other processes" = raised beach crest degradation and wetland plain degradation, or boulders; *Unknown proces*s = unclassified. Biotic processes = Vegetation succession (succ.) or aggradation (aggr.). See table 1 for more detailed descriptions of the processes.**





## 6.7    Appendix G

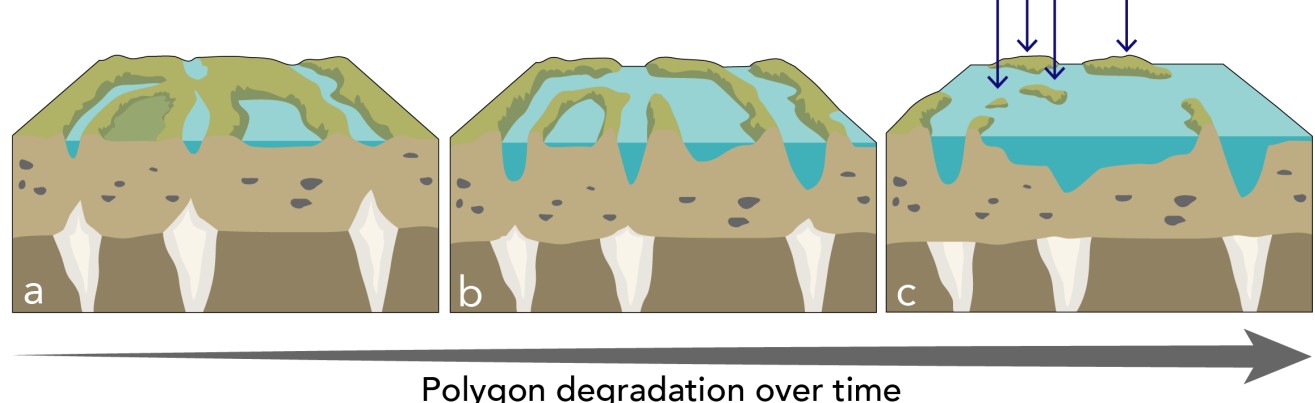

**Figure G1. Schematic representation of the polygon degradation process in wetlands (a-b), generating islets in (c), pinpointed by the blue arrows. These islets have a wide range of physical characteristics (e.g., distance to shore or water depth).**

## 7    Data/Code availability

Complete data associated with this article can be found online https://doi.org/10.5281/zenodo.8395558.

## 8    Author contribution

**Madeleine-Zoé Corbeil-Robitaille :** Conceptualization, Methodology, Data curation (equal), Formal analysis (lead), Investigation, Writing – Original draft (lead), Visualization. **Éliane Duchesne** : Methodology, Data curation (equal), Formal analysis, Investigation, Writing – Review & Editing. **Daniel Fortier:** Methodology, Formal analysis, Writing – Review & Editing. **Christophe Kinnard :** Writing – Review & Editing. **Joël Bêty :** Conceptualization, Methodology, Investigation, Writing – Review & Editing, Supervision.

## 9    Competing interests

The authors declare that they have no conflict of interest.

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
