# Peer review of "Linking biodiversity and geodiversity: Arctic-nesting birds select refuges generated by permafrost degradation"

_EGUsphere, 2023_

## Author Response (AR1)

**Letter to reviewers**
*Linking geomorphological processes and wildlife micro-habitat selection: nesting birds select refuges generated by permafrost degradation in the Arctic* (egusphere-2023-2240)
Madeleine-Zoé Corbeil-Robitaille, Éliane Duchesne, Daniel Fortier, Christophe Kinnard, Joël Bêty
*Biogeosciences*
*Received: 30 Sep 2023; Discussion started: 12 Oct 2023; Discussion ended: 8 Dec 2023*

Object: Final response

Dear Reviewers,

Thank you for your insightful and constructive feedback on our manuscript now re-titled "***Linking geomorphological processes and wildlife micro-habitat selection: nesting birds select refuges generated by permafrost degradation in the Arctic (egusphere-2023-2240)***". We appreciate the time and effort you've dedicated to reviewing our work. Below, we provide responses to your comments and suggestions.

Our corrections firstly address concerns regarding the clarity of terms, alignment of title with content, systematic presentation of data. We also address concerns regarding the analytical approach, the interpretation, and refined some discussion points, like findings related to bird habitat selection and the influence of environmental factors.

We believe these revisions enhance the clarity and completeness of our study. We are grateful for your valuable feedback.

Sincerely,
Madeleine-Zoé Corbeil-Robitaille et al.,

*Linking geomorphological processes and wildlife micro-habitat selection: nesting birds select refuges generated by permafrost degradation in the Arctic* (egusphere-2023-2240)

**RESPONSE TO REVIEWER #1**

**General comments**

A very nice article reflecting both interesting and hitherto not fully explored topic. The manuscript is overall well written and interesting, and I did not find any major flaws or inconsistencies. I have few comments for your consideration.
First of all, I especially appreciate extensive fieldwork behind this manuscript. Empirical data is very much needed to gain knowledge of our Arctic landscapes. Well done with that!
      **RESPONSE: We would like to thank you for your comments on the manuscript. We're glad you enjoyed reading it and are glad to share your views on the need to gain a better understanding of the Arctic landscape through in-depth fieldwork.**

**Specific comments**

Main issue is about the use of the term geodiversity in the Title, Introduction, and Discussion and whether it is used clearly. Geodiversity consists of geological, geomorphological, and hydrological variation of the earth's surface and subsurface (Gray 2013). I think you should sharpen the message of the manuscript especially in aforementioned sections as you do not assess or use the geodiversity or biodiversity through species richness or georichness, respectively, but rather have a case study of how certain aspects or features of geodiversity (here polygon degradation, glacial boulders, and raised beach crest degradation) are linked to arctic-nesting birds. So, I would see your approach to geodiversity is qualitative, through certain geomorphological features or landforms like f.e. Tukiainen et al. 2019 has done in the Journal of Biogeography.

The first paragraph (Starting from L 34), is about geodiversity and its relevance to the living world. Firstly, please add of what things geodiversity consists of (See Gray 2013). In addition, it should be initialized what kind of approach this manuscript is taking, that isa the qualitative approach to geodiversity-biodiversity relationships.
      **RESPONSE: This comment is very relevant. Following your suggestion, we changed the title and added to the manuscript a short description of geodiversity following the definition proposed by Gray. We have also specified the approach used to study the links between geodiversity and biodiversity and clarified the focus of our study following your advice:** *"In this study, we use a qualitative approach to investigate Arctic geodiversity-biodiversity relationships by assessing how certain geomorphological features may be linked to Arctic birds nest selection."*

On page 13 L 326 you describe what you have done: linking geomorphological processes and wildlife micro-habitat selection. I would reconsider the title of the manuscript to better fit with the contents of the manuscript f.e. by dropping off the holistic terms geodiversity and biodiversity and adding something more specific f.e. "linking geomorphological processes and wildlife micro-habitat selection". Geodiversity would fit greatly into keywords of this manuscript.
      **RESPONSE: We totally agree and modified the title.**

In Table 1: To emphasize geodiversity, please specify which islets are considered as a part of geodiversity and what is not (biotic process one).
      **RESPONSE: We have adjusted table 1 according to your suggestions.**

I think these results contribute to our knowledge about Arctic environment and these kinds of studies that bring empirical evidence about the relationship between abiotic and biotic nature are very much needed.
Did you consider adding any other variables into your analyses?
      **RESPONSE: Yes. Indeed, many variables can potentially affect habitat selection in birds. As indicated in the Discussion section, "***Nest site selection can be influenced by several factors that were not considered in our study. For example, site selection by Red-throated loons can depend on lake or pond characteristics (e.g. bottom topography, looseness of pond floor, distance to the ocean (Douglas and Reimchen, 1988; Eberl, 1993)). Adding such variables to our analyses would likely improve our ability to explain the probability of nest occurrence on islets***".**
**It is challenging to obtain enough data to fully explore the combined influence of several variables on nest selection by arctic birds. Considering that nest predation is the main cause of nest failure in our study area, we decided to focus on characteristics that can impede Arctic fox movement and tested a well-defined a priori hypothesis. We didn't have the data to consider vegetation or substrate type. Although it was based on a sub-sample, we were able to explore the effect of lake and islet areas on selection. Adding these two variables did not alter our main conclusions.**

**Technical corrections**

Like said earlier, I find the text easy to follow for a reader not so familiar with birds and I didn't spot any grammatical errors.

In the appendix D. please present each species' tables systematically in the same order than in the manuscript figure 4. (1st Glaucous gull, 2nd Cackling goose and 3rd Red-throated loon).

**RESPONSE: We standardized the order of named species throughout the manuscript (following 1st Cackling goose, 2nd Glaucous gull and 3rd Red-throated loon) and hence modified the figure 4.**

*Linking geomorphological processes and wildlife micro-habitat selection: nesting birds select refuges generated by permafrost degradation in the Arctic* (egusphere-2023-2240)

**RESPONSE TO REVIEWER #2**

**General comments:**
The paper assesses the physical characteristics of small islets in Arctic environments. These islets often serve as nesting platforms for birds, and the authors nicely document the physical characteristics of islands that were selected for nesting compared to unoccupied islands. In a helpful subsequent step, the authors next characterized the geomorphological process by which each islet was formed, finding that ice-wedge polygonal degradation was the primary genesis of islets at their study site. Together these assessments provide a useful overview of the factors that promote the formation of islets and their occupancy by nesting bird. The figures are informative and easy to understand, and the authors employed appropriate analytical approaches to address their study questions. The paper was well written and interesting to read, and the authors provide good context for their findings and discuss the role of climate change in future creation and degradation of Arctic islets. I had very minor suggestions on rewording, syntax, etc., but more substantive suggestions for the authors on ways to improve their analysis of physical factors that promote the occupancy of islets. I hope the authors find my comments to be helpful.
Dan Ruthrauff
US Geological Survey Alaska Science Center
druthrauff@usgs.gov

       **RESPONSE: Thank you for your thoughtful feedback. We took them all into consideration and have taken the time to respond. Please note that we have considered all technical corrections (e.g. "image" to "imagery"). Simple corrections are not mentioned below but have been made in the manuscript.**

**Specific comments:**
In general, your methods and analyses are appropriate for your questions and are clearly presented. I do think, however, that your manuscript would benefit from a more straightforward analytical approach regarding your assessment of factors that promote islet occupancy. For this, you essentially have two model sets, one including measures of islet area and lake area, and one without. You go to a lot of trouble to show results from both sets, which I found a bit confusing…but ultimately base your inference on the model set without measures of area as covariates. You state that your findings regading Distance and Depth do not change with Area as a covariate…which to me begs the question of why you then bother excluding Area? You state that this was due to sample size concerns (n=315 islets with all measures, n=350 with distance and depth), but both model sets employ pretty robust sample sizes. Since islet and lake area seem like biologically relevant covariates, I'd just stick with your 'larger' analysis, and not re-run models with area removed. Having the two model sets creates confusion between Table 2 and Appendix 3. I also had some questions about the models in your model set. Unless I'm mistaken, you did not create models that did not include either DISTANCE or DEPTH as a covariate (except for the null model). Having models in your model set with only IsletArea, LakeArea, and an additive model using these same covariates would help better assess the influence of the areal measures on islet occupancy.
       **RESPONSE: As suggested, we ran the proposed models, adding combinations that include surfaces only (we've replaced the tables in the appendix). We have included surface parameters in our model selection and provide all the details in the Appendix. Our conclusions remain the same. As indicated, two variables (DISTANCE and DEPTH) are the ones we aim to focus on, as we hypothesize that these characteristics can impede Arctic fox movement. We revised the text to clearly indicate that we focus on these two variables. As indicated in the manuscript, our dataset is reduced when including surface parameters (from 350 islets to 315). The islets removed are mainly located close-to-shore, where we detect the strongest effect on nest occurrence probability. By adding 35 islets to the dataset, we add 32 islets in the "0-10m" distance category. This affects the selected distance weighted functions, and we thus prefer to use the best dataset to test the a priori hypothesis and to illustrate the effect of DISTANCE/DEPTH on nest occurrence. This being said, we agree that other variables complement our work, and we now provide all the results (including parameter estimates) obtained using a smaller sample size in the Appendix.**

\*\*\* Also, you say that models within ≤2 ΔAICc were 'considered', but there's no sign what this actually means. You present only the parameter estimates from your top model, so it doesn't look like other well-supported models were considered. Given that most of your outputs had pretty equivocal model support, I think you should consider model averaging to estimate parameters. Selecting only the top model is generally not well supported, especially when there's high model uncertainty.
       **RESPONSE (*to the present comment and comments Line 153, Line 160, Lines 188-190, Line 422)*: We think that the use of model averaging is not appropriate in our study. We compared models with or without distance-weighted functions for the same variable (DISTANCE and DEPTH). We cannot use model averaging in such case. Our top models generally differ from the best fitted model based on the presence/absence of these functions. In that**

context, it is not surprising that the best-supported model does not have overwhelming model weight. Note also that model averaging is especially relevant when the focus of the study is around prediction. This was not our primary goal as we wanted to test the hypothesis that birds select islets less easily accessible by Arctic foxes (i.e., those farther from the shore and surrounded by deeper water). Despite of some uncertainty in model selection, we found very strong support for an effect of Distance and/or Depth, and hence our results strongly support our hypothesis. To visualize our results, we used the coefficients of the best supported model for our 350 islets dataset.

We made a correction in the text "*We considered models with an AICc less than or equal to 2 to be competitive. Coefficients of the best-supported model were used to visualize the results*".

We have revised the overall organization of Appendix D to avoid confusion. Additionally, we have included models containing only surfaces in the model selection.

We also modified Table 2 to include all models with AICc ≤ 2.

**Technical corrections:**
Line 20: what is journal format regarding adding genus and species names at first mention of a species?
      **RESPONSE: This doesn't seem to be specified in the journal guidelines. Latin names are listed at first mention in the introduction.**

Line 27: 'linearly or nonlinearly' is confusing; as the reader does not yet know about your distance-weighted function, I'd just drop 'linearly or nonlinearly' from the abstract. The truth of the statement remains intact. Also, 'and/or' is more clearly just 'and'.
      **RESPONSE: We followed your advice.**

Line 52: Caro missing year?
      **RESPONSE: We adjusted the bibliography!**

Line 144: not sure you need to mention that you didn't fit random effects…I guess I only mention it if I do fit random effects.
      **RESPONSE: Probably not, indeed. We removed this information.**

Line 153: change 'lesser' to 'less'. Also, I'll await results, but when you say models with deltaAICc ≤2 were considered, how did you consider them? Model averaging? OK, having read more thoroughly, it seems that you only show parameter estimates from your top model (Table 2). In this sense, I'm not sure how you 'considered' the other models? A real advantage of AIC modeling is the ability to conduct model averaging for drawing inference; generally, drawing model inference from the best-supported model alone is poorly supported, unless it has overwhelming model weight (which yours do not).
      **RESPONSE: See response to comment above (*\*\*\*Also, you say that models within ≤2 ΔAICc were 'considered' [...]*).**

Line 160: so, this is a bit unclear. You present 'full' model results in Appendix D, but here state that you removed LakeArea and IsletArea due to missing data. One idea to consider is that if you didn't include these two measures of area in your final modeling, then you should not mention them at all in the paper. Alternatively, despite the smaller sample sizes, since your model results don't differ when you do include these area-related variables, I'd probably just keep them in the paper—these seem like biologically relevant measures, even if not collected at all sites. Readers like me would probably wonder about the effects of the lake size and islet size. As it stands, you introduce them and then remove them. I'd advocate for just including them so you can more fully discuss them. But, note that due to the removal of area measures, Appendix D is not really comparable at all to results in Table 2. The AICc values and weights are totally different…so, it's really an apples-to-oranges comparison to have both. They are different model sets, and not comparable; you sort of walk a middle path between the two sets, which I found confusing. I think it would be clearer were you to base all your results on the 'full' results from Appendix D rather than the subset in Table 2.
      **RESPONSE: See response to comment above (*\*\*\*Also, you say that models within ≤2 ΔAICc were 'considered' [...]*).**

*Linking geomorphological processes and wildlife micro-habitat selection: nesting birds select refuges generated by permafrost degradation in the Arctic* (egusphere-2023-2240)

Line 183: 'best fitted' implies some measure of actual fit…so I prefer to use terms like 'best supported' in AIC modeling frameworks. This term does not imply that the model is actually 'good', only that it's the best supported—it's a more neutral way to frame it.

**RESPONSE: We totally agree and have used "best supported" instead.**

Lines 188-190: see comments above re. including area. N = 315 is still a pretty robust sample. So, including lake and islet area didn't really change the relationship between nest occurrence and distance and depth…but what were the relationships to area? As I mention above, I think you've got a nice sample size, and restricting your analysis to only islets where you had distance and depth gains you n=35, right (315 v. 350). I'd keep area in your models and discuss this effect. Also, it looks like you didn't include any models in your model set that did not include either distance or depth (other than your null model and a spatial model)? Why did you not include models with IsletArea, LakeArea, and IsletArea + LakeArea (ie, Areas) alone as models? Seems you haven't really assessed the influence of area without such models. I see on line 380 you summarize these results (occupied islets tend to have greater IsletArea and LakeArea than unoccupied), but this is not in the main results. I'm also confused why results in Table 2 don't mimic those in Appendix Dx.2? For instance, in Table 2 for CACG you show DISTANCE* + DEPTH (w=0.2) and DEPTH (w=0.26). In D1.2, which should be the same as what's presented in Table 2, you show the 2 aforementioned models but also two more models within deltaAICc of 2. Why were the other two models in D1.2 (DISTANCE + DEPTH, DISTANCE* + DEPTH*) not shown in Table 2?

**RESPONSE: See response to comment above (*\*\*\*Also, you say that models within ≤2 ΔAICc were 'considered' [...]*).**

Line 200: nice figure! This clearly shows the relationship between depth and distance across used sites for each species. Also, you previously ordered species in results as CACG, GLGU, RTLO, but here it's GLGU, CACG, RTLO, might swap them around to maintain order throughout.

**RESPONSE: Thanks! Lot of work on this figure. As we answered to Rev#1, who had the same inquiry, we standardized the order of named species throughout the manuscript (following 1st Cackling goose, 2nd Glaucous gull and 3rd Red-throated loon).**

Line 240: nest site selectin by loons varies by loon species. In Alaska at least, RTLO breed on small ponds not otherwise occupied by PALO or YBLO. These ponds typically freeze deeply in the winter, so RTLO typically feed in the marine environment. PALO and YBLO, in contrast, nest on deeper lakes with more abundant food resources. Most chick provisioning occurs from within the nest lake itself for PALO and YBLO. So, for 'loons', food availability is also a factor in site selection. For RLTOs specifically, this is probably not the case, so you may want to explicitly state 'red-throated loons' here instead of 'loons' more generally.

**RESPONSE: This is relevant – we specified the species throughout the manuscript.**

Line 333: this reads as if the primary way that climate change alters predator-prey interactions and the occurrence and distributions of species in the Arctic is via influencing refuge availability through ice-wedge degradation. Of course, climate change is rapidly and markedly changing predator-prey interactions and the occurrence and distributions of species in the Arctic…but via a multitude of mechanisms, not just ice-wedge degradation. Consider rewording: you're really just trying to say that climate change is irrevocably altering Arctic ecosystems, and the role of climate change effects on ice-wedge degradation and how this relates to predator-free nest sites had been previously little studied.

**RESPONSE: We used your suggestion and reworded it to nuance the passage** *"Climate change is irrevocably altering Arctic ecosystems through multiple mechanisms. Its effects on ice-wedge degradation and their relationship with nest site selection by birds had been little studied before. Given its influence on refuge availability through ice-wedge polygon degradation, islet formation and changes in islets topography over time, we can reasonably conclude that global warming is likely to alter predator-prey interactions, species occurrence and distribution in the Arctic landscape."*

Line 384: 'It seemed adequate to work this way with our variables' is a bit underwhelming. Maybe restate this more positively and assuredly? 'Based on our hypotheses about the effects of distance from shore and islet depth on site use, distance weighted functions provided an appropriate model framework for our data structure' or something similar.

**RESPONSE: We used your suggestion as is.**

Line 422: as stated previously, there's not much support for just presenting the results of the model with the smallest AICc, especially when the model support is pretty equivocal. Why do you not model average? I think you'll need to state why your opted not to implement model averaging.

**RESPONSE: See response to comment above** *(\*\*\*Also, you say that models within ≤2 ΔAICc were 'considered' [...]).*

Appendix E2: this is interesting! I'd love to know about successful vs. unsuccessful nests. At our study site in northern Alaska, the depth of the water around the islets is much greater, but almost without fail these deep-water islands are visiting by swimming foxes who depredate all the nests on the islands (typically black brant nests). In comparison, the depths you measured at your site are quite shallow compared to the sites that I'm familiar with in Alaska.

**RESPONSE: Gauthier et al. 2015 (cited in our paper) found that the hatching success of glaucous gulls was greater for nests on islets than at the shore in our study area (info added in the methods). We are currently investigating the effect of islet characteristics on predation risk using both artificial nests and long-term nest monitoring of gulls and cackling geese. This is the focus of another MSc thesis, and the results will be integrated into another manuscript. Although we agree that adding information on predation rate would complement our study, we think that our paper is already providing a large amount of new and original data. Note that we are also currently investigating the effect of prey density in the landscape on the predation risk for birds nesting on islets. We have some evidence suggesting that foxes are more willing to visit islets when their prey acquisition rate (more specifically their energy acquisition rate) is below a certain threshold. That may explain some annual and spatial (inter-site) variation observed in the arctic tundra. We hope to publish these exciting results in the near future.**

*Linking geomorphological processes and wildlife micro-habitat selection: nesting birds select refuges generated by permafrost degradation in the Arctic* (egusphere-2023-2240)

**RESPONSE TO REVIEWER #3**

Dear authors,
It was interesting to read your study, which has an interesting view angle (relation between both types of diversity) and I find well written. I have several main points and more detailed point are given below.

General comments

**General response to Reviewer 3: Thank you for providing your valuable feedback. We have carefully reviewed your suggestions and modified our manuscript. While we have addressed technical corrections, we may not explicitly mention minor corrections. Your input has been thoroughly considered, and we greatly appreciate the time you took to provide it.**

(1) To me it is unclear to what extent polygon degradation is a cyclical process (as you say. A long-term process) or sped up by climate change (a short term process via permafrost breakdown). You start from the climate change perspective, but the time scale and relative contribution of processes is unclear. Related to this, if the process of permafrost/polygon degradation is warming-induced, what was then the historic distribution/habitat choice of the species in the landscape? Was it forced to nest in more accessible locations in the past due to absence of distant islets? Is that not the case anymore now?

 **RESPONSE: Polygon degradation may be part of a long-term process, notably linked to site-specific conditions such as thicker snow, water ponding and run-off, and thermal-erosion for instance. In this sense it is part of the long-term evolution of polygonal landscapes. Polygon degradation can also occur due to climate changes (e.g. atmospheric temperature change, precipitations changes) or to climate extremes such as a significantly warmer summer.**

 **We can only comment on the distribution of species in recent years, as we started to focus on the three study species relatively recently. We do not have historical data and agree that it would be very relevant to better understand the current patterns and to anticipate futur changes. According to our recent monitoring, the study species predominantly nest on islets, but it is possible to find some nests on the shore. These nests are more vulnerable to predation (Gulls: Gauthier et al. 2015). We lack data to support what might have occurred before monitoring, but we can hypothesize that the warming-induced increase in rate of polygon degradation could impact the availability of nesting sites in the future.**

(2) please consider the issue that you seemed to be unable to include failed nests in your analysis. This has important effects on the conclusions that you can draw.

 **RESPONSE: Every year, we intensively search for nests early in the incubation period, but it's possible that we haven't been able to locate all the early-failed nests. In most years, nest predation risk is relatively low for birds nesting on islets (Gauthier et al. 2015). The hatching success of gulls nesting on islets is typically between 90-100%. This reduces the potential effect of undetected "failed" nests on the observed patterns. If we were unable to include failed nests, it could have artificially increased the probability of nests occurrence on low-risk islets. However, we believe that this is relatively rare, since the nest cups of all three species monitored are built and tended. If the nest fails, we can find a nest-cup with a well-defined structure containing fresh material such as down in the case of Cackling geese, vegetation and feathers in the case of Glaucous gulls, and wet vegetation in the case of Red-throated loons. We can also find fresh egg shells in depredated nests. Although it was rare, the presence of egg shells and fresh nest-cups were considered in our analyses. We added the information in our methods.**

**We added these sentences in the manuscript to clarify:** "*In most years, nest predation is low for birds nesting on islets on Bylot Island (Gauthier et al. 2015). Occurrence was also assessed by the presence of fresh nest material and eggshells found in empty nest cups. Although we may have missed a few early-failed nests during our visits, we are confident that the vast majority of unoccupied islets (i.e. no nest was found over the two-year study period) were not used by nesting birds.*"

(3) please also consider possible effects of bird species on each other.

**RESPONSE (also to comment L190): This is a good point, and we have now acknowledged such potential effect in the methods. The species investigated in our study system can nest very close to each other. All 3 species can nest on the same islet. The shortest interspecific distances recorded in the field are 1m between a goose and a loon, 1m between a gull and a loon, and 9m between a goose and a gull. Of course, the few very small islets cannot be occupied by more than one species due to their size: they therefore become unavailable to others once occupied. However, the proportion of occupied islets is relatively low (24%) in our study area, and we can assume that the occupation of the very small islets by a given species did not prevent another species from selecting islets with similar distance to shore and water depth (the targeted physical characteristics in our study). We thus assumed that it did not affect our ability to investigate the effect of islets characteristics (DISTANCE and DEPTH) on the probability of nest occurrence.**

**Sentences added:** "*The presence of a bird species on an islet may influence the likelihood of finding another species on the same islet. We did not consider such inter-specific interactions in our study, and we assumed that it did not affect our ability to investigate the effect of islets characteristics (DISTANCE and DEPTH) on the probability of nest occurrence. This assumption is likely valid because i) the proportion of occupied islets is relatively low in the study area (24%), and ii) the study species can be found on the same islet and can nest very close to each other (minimum distances between nests: 1m between loons and gulls, 1m between loons and geese, 9m between gulls and geese). Although some very small islets could not be occupied by more than one (or two) species, the relatively high availability of unoccupied islets in the landscape likely allowed most birds to use islets with the preferred characteristics.*"

**TITLE**
It is not clear that they select them for nesting.
Also, your article is less concretely linking both types of diversity. You study where birds nest. Maybe the reader expects analysis of a correlation between both types of diversity (comparing areas).

**RESPONSE: We modified the title to clarify according to your comments.**

Also, do you mean permafrost degradation, or polygon degradation? What have you shown?

**RESPONSE: Polygon degradation is inherently a form of permafrost degradation. Conversely, permafrost degradation manifests through various types of changes such as polygon degradation, thaw slumps, thermokarst subsidence, thermokarst ponds/lakes, etc. Hence, permafrost-related geomorphological processes include polygon degradation. We prefer to use a broad term that is easily understood by readers from various disciplines, including biologists.**

**METHODS:**
Table 1: since you introduce this topic from the climate change background, it would be good to know which processes are affected by climatic warming. Only 1-2?

**RESPONSE: 4 out of 6 processes may be affected by climate warming: 1) Low-center polygon degrading in ridge-like islet, 2) flat or high-center polygon degrading in center-like islet, 5) water level variation (i.e. evaporation in warm years/low precipitation) and 6) vegetation aggradation (via growing season in warm/cold years + wetness/dryness during warm/cold/humid/dry summers). We mentioned in the Discussion the potential impact of climate change on the main geomorphological process generating islets in the study area.**

Table 1: can the 2 processes in category 6 be distinguished?

**RESPONSE: No, unfortunately not. The two processes create islets that look similar on satellite imagery, and the field identification was not detailed enough to distinguish between them.**

Table 1: can you give an indication of the timescale of the processes?

**RESPONSE: This is a good point. We do not have data to provide the exact timescale of each process in the study area. Time scale is also climate dependent. We thus prefer to avoid adding such information in Table 1. We added information on timescales in the Discussion (see section 4.3) and in the Appendix (Figure G1).**

*Linking geomorphological processes and wildlife micro-habitat selection: nesting birds select refuges generated by permafrost degradation in the Arctic* (egusphere-2023-2240)

97-99: this is about the historic situation, but this may change if the prey nests more and more on islands, where foxes cannot predate them. Any data / discussion on that?

**RESPONSE: Foxes in our study system can prey upon nests located on islets. We suggest that nesting on islets may reduce the probability of predator encounters (less accessible to foxes), as indicated in the abstract, introduction and discussion.**

**Avian predation generally results in partial clutch predation, involving the removal of part of the clutch (e.g., Bêty et al, 2002), whereas fox predation is generally total (Beardsell et al, 2021). Since the beginning of long-term monitoring on Bylot Island, foxes have remained the most important nest predator in the system, but their relative importance may be lower for species nesting on islets. We have no data to fully support that statement and we are currently investigating the effects of islets characteristics on predation risk (using artificial nests, nest monitoring and automatic cameras). Note that foxes can find many preys in the study area that are not on islets (including lemmings, shorebirds, snow geese, passerines, jaegers). As indicated in the methods, only 3 of the 35 bird species essentially nest on islets.**

119-122: it is important to know whether you were able to detect failed nests. If not, then your analysis of the nest site selection of prey bird species, may in fact not show prey nest site selection, but predator predation success: they removed all nests on easily accessible locations, which led you to conclude that prey only nests on safe locations far away. The subsetting to active nests is tricky in your analysis.

**RESPONSE: See above – response to *General comment (2)* for more information on this topic.**

119-122: also, how were they observed? How closely did you inspect the islets? Did you step on them to check for sure?

**RESPONSE: We added the following sentence to clarify: "*We visited islets, stepping on each while taking measurements.*"**

119-122: what about a situation where you can have species A in year 1, species B in year 2?

**RESPONSE: The occurrence was determined by species. 84 islets were occupied at least once by at least one species, which does not exclude the possibility that certain islets were occupied in both years by the same species or two different species. See also above – response to *General comment (3)* for more information on this topic.**

133: what about a situation where an islet is in fact the result of a combination of processes? I would guess that is quite common?

**RESPONSE: Based on our definitions and criteria, a given islet cannot be associated to more than one process. However, biotic-related processes could accelerate the formation of islets e.g. Loons may accumulate vegetation on well degraded polygons (submerged). In that case, the process generating the islet was identified as vegetation aggradation.**

134: how reliable is that method? Do you have sources/references? Was that validated by soil analysis? Was the categorisation done by one same person?

**RESPONSE: Our method is valid and recognized in geomorphology study, remote sensing and satellite image interpretation. The classification was mainly performed by the first author and validated by co-author Prof. Daniel Fortier, an arctic geomorphologist that has worked on Bylot Island since 1999. It has been done without prior knowledge of where birds nested. We provide enough details in our paper to replicate our study and we encourage other researchers to improve the methods we developed.**

**The method we used is based on the characteristics of the islets. These characteristics can be geomorphological or biological and they refer to published research on the Quaternary, periglacial geomorphology, and plant ecology of Bylot Island and elsewhere in the Arctic. This classification was designed for the study area but could be adapted and exported to other sites. Polygonal landforms, either intact or degraded, are usually easily detectable on aerial pictures and satellite imagery. They occur in various types of surficial sediments, so a soil analysis is not required to validate the interpretation. Nevertheless, the sites/islets were all visited in the field to confirm the interpretation. We have also consulted various field experts (e.g., Prof. Esther Levesque, Dr Samuel Gagnon, MSc Alexis Robitaille and MSc Karine Rioux) and gathered information from the literature on various processes generating landforms in the Arctic prior to defining how to classify islets. The visual features used for classification are derived from published works.**

Did you test how consistent the categorisation method is by letting multiple people assign them and see if they come up with the same result? Was the classification done without prior knowledge of where birds nested?

**RESPONSE: We did not. As indicated above, the classification was mainly performed by the first author and validated by co-author Prof. Daniel Fortier, an arctic geomorphologist that has worked on Bylot Island since 1999. We have used "unknown process" in cases of uncertainty, and 328 out of 396 islets (83%) were associated with a specific geomorphological or biotic process with high level of confidence. Our main conclusion is thus robust (i.e., ice-wedge polygon degradation generated most islets in the landscape). We provide enough details in our paper to replicate our study.**

144: When analysing selection, you assume that birds have something to choose. Is that the case? Is the area not saturated? How strong is the competition for good islets?

**RESPONSE: No, the islets are not saturated. As indicated in the results, a total of 84 islets out of 350 (24%) were occupied by a nesting bird (Cackling goose, Glaucous gull, or Red-throated loon) at least once during the study period. If we consider the 2-year occupancy, 97 islets out of a total of 396 available are occupied by one species, sometimes by two, which is equivalent to about one islet out of 5 being occupied (of these, 84 out of 350 islets for which we had distance and depth were occupied during the study period). Many islets are still unoccupied in the study area. Note that the cackling goose population is currently increasing (exponential growth) and we anticipate that competition will increase soon. We plan to document the impacts through long-term monitoring.**

151: explain that only additive models were included. Why not interaction effects?

**RESPONSE: We used declining distance functions to transform our main variables of interest. This captures potential interactions between main characteristics (DISTANCE and DEPTH). By testing models with and without declining distance functions, and by selecting the best fitting decay distance function to transform the DISTANCE and DEPTH according to their declining effect, we can detect relevant interactions between the main variables. For instance, as illustrated in Figure 4a2, DEPTH does not have an impact on the probability of occurrence of Glaucous gull when DISTANCE is <5 m; above 5m, the effect of DEPTH becomes stronger.**

156: including just lat+lon is a simple method, which I think is not accurate and does not account for clustering (especially if more clusters occur). It only works if there is 1 cluster which is in the extreme of lon/lat. It does not work when there are multiple clusters spread out over the map, or 1 in the middle of the map. Can you think of a better method, e.g. internest-distances compared to a random situation, or presence of hetero-/conspecifics around compared to a random situation?

**RESPONSE: We changed the wording, as we wanted to consider "spatial correlation", and not "clustering". In the present case, adding coordinates as predictors in statistical models can account for spatial correlation to some extent. By including geographic coordinates as predictors, the model can account for spatial trends that may exist in the data. We did not capture any trend.**

**RESULTS:**
166: >41: why no exact value?

**RESPONSE: The reason lies in fieldwork constraints/efficiency combined with our main hypothesis. The leg length of the Arctic fox is around 30 cm, hence above that, fox would have to swim to reach an islet. We measured depths of up to 40 cm in the field using graduated sticks, and all measurements above this value were grouped into one category, as it would not change the accessibility to foxes.**

167: 84 islets were occupied, but what if species A in year 1, species B in year 2?

**RESPONSE: See above – response to *L119-122 and General comment (3)*.**

169: why not compare them with unoccupied islets? (instead of all available)

**RESPONSE: We wanted to compare the selected islets to the entire set of available islets (including those that were occupied). This is a common permutation test performed in habitat selection study as it allows the identification of preferred habitat characteristics considering what is available in the landscape.**

Figure 3: the categories in panel b combine 2 variables with parallel gradients. However, in L. 116 it is said that the variables are not correlated. This is confusing and I cannot evaluate whether the categories in panel b make sense (they would only when there is a correlation between these 2 variables).

**RESPONSE: Figure 3 is descriptive, and panel b only shows the distribution of islets across the study area considering the water depth and distance to shore, divided into 6 categories to make the visualization easier on the map. The reader can quickly get information on the distribution of islets, at the landscape scale, over the gradient of characteristics observed in the field. As shown in Figure 3, some islets can be close to shore but surrounded by high water depth (dark blue dots), while some can be far from the shore and also surrounded by high water depth (dark red dots). Figure 4 provides the number of islets for each combination (DISTANCE and DEPTH) but using higher precision. The correlation between these variables was relatively weak (we slightly modified the text in the methods – section 2.2).**

188 (and also L. 160): should be in methods. The cases with missing data are confusing: you should just be able to look at a satellite image. Or were those island too small? Then you have missing data especially for 1 category (i.e. very small islands, vegetation aggradation/loon).

**RESPONSE: We slightly changed the method and result sections to avoid confusion. We specified in methods that "*We georeferenced islets in the study area using a combination of satellite image analyses and intensive field surveys conducted during the bird incubation period*". The majority of islets have been pinpointed by satellite imagery before fieldwork. During fieldwork, we visited each of them, then visited every lake in our study area and took coordinates for each new islet found. Afterwards, we measured surfaces by imagery. Even though our images have a fine resolution (0,3m), some islets were not visible on the images, which precluded our ability to estimate surface characteristics.**

**We may have underestimated the number of islets in the "very small islets category" but have nuanced it in the Discussion 4.3: "*Biotic processes such as vegetation aggradation or succession are the second most common processes that generated islets in the study area (about 10% of those that could be classified). We may have slightly underestimated the number of islets associated to this category, as they are generally smaller and perhaps harder to interpret in the field or to classify using satellite images*".**

190: It would be good if you could also test the effect of species on each other. If one present, than others around? Hetero/conspecific clustering? And can you describe your data/situation better? I.e. are there more nests/islets in the same pond? (maybe for gulls/geese that is true but not loons, which forage in the pond). How many? Do different species occur in the same pond? Can you give some data about this? Inter-nest distances between con/heterospecifics? Never multiple nests on one islet? Data? Which species?

**RESPONSE: See response to General comment (3).**

197 (Table 2): so the distance-weighing is because the difference between 101 and 110 m is hypothesised to be less important for the bird than the difference between 1 and 10 meter? Is it not better to log-transform the units distance and depth? Then, it is easier to compare/interpret the effect sizes between species when you have the model output.

**RESPONSE: We agree that different approaches could be used. The declining distance method is widely used in ecology and habitat selection, providing a comprehensive understanding of the diminishing effect of one variable on another, as mentioned in the Appendix C ("*Distance weighted functions such as a negative exponential function paired to a distance function enable the consideration of the continuously declining effect* [therefore not linear] *of the surrounding landscape on an ecological response with increasing distance from the point where the response is measured (Miguet et al., 2017)*"). The use of this method is justified and appropriate in our study to describe the influence of physical parameters of an islet on nest-site selection.**

**DISCUSSION:**
223: coexistence of which species? Predator and prey?

**RESPONSE: Coexistence of different prey that would be otherwise excluded by predator-mediated effects (Holt 1987). We specified "*prey*" in the manuscript.**

227: 3 species: but do they breed at the same time? Do they compete and does one species displace the other? In what order do they arrive and breed, and displace each other?

**RESPONSE: See our response above. As mentioned, more than three-quarters of the islets in our study area were not occupied. We also have recurrent observations of 2 or 3 species nesting on the same islet. At Bylot Island, Glaucous gulls usually nest first, followed by Cackling geese and then Red-throated loons, but nest initiation periods overlap. We have no indication that one species can displace another once established.**

230: Do those islets also become snowfree earliest? Were your years 2018-2019 late or early snowmelt years? Same about the predation: Were your years 2018-2019 low or high predation years? And in the quantitative sense, what was the predation pressure (e.g. foxes per square kilometer)?

**RESPONSE: We do not have data on snowfree dates at the scale of islets. At the landscape scale, snowmelt in 2018 was average whereas 2019 was considered early. As indicated in the paper, in most years nest predation is low for birds nesting on islets on Bylot Island (Gauthier et al. 2015). The year 2018-2019 were typical years. We are currently investigating the effect of islet characteristics on predation risk using both artificial nests and long-term nest monitoring. This is the focus of another MSc thesis, and the results will be integrated into another manuscript. We are also currently investigating the effect of prey density in the landscape on the predation risk for birds nesting on islets. Although we agree that adding information on predation rate would complement our study, we think that our paper is already providing a large amount of new and original data. Our main conclusions are robust and supported by high quality data.**

232: what did that study do?

**RESPONSE: We specified what Eveillard-Buchoux et al. 2019 did in their study:** *"(e.g. (Eveillard-Buchoux et al., 2019) linking nest-site geomorphology to cliff-nesting species preference)"*

241: what about food availability?

**RESPONSE (As answered to Rev#1): The availability of resources can be an important factor affecting habitat selection in birds. As indicated in the Discussion section, "***Nest site selection can be influenced by several factors that were not considered in our study. [...] Adding such variables to our analyses would likely improve our ability to explain the probability of nest occurrence on islets ***". It is challenging to obtain enough data to fully explore the combined influence of several variables on nest selection by arctic birds. Considering that nest predation is the main cause of nest failure in our study area, we decided to focus on characteristics that can impede Arctic fox movement and tested a well-defined a priori hypothesis. As indicated above, we are also currently investigating the effect of prey density in the landscape on the predation risk for birds nesting on islets.**

243: but how much interannual variation is there? And how large could the effect possibly be?

**RESPONSE: As mentioned in Discussion (4.1 Physical characteristics and nest site selection), this is one of the limitations of our study: we don't have the data to answer this question. It would be interesting to take measures of distance and depth every year and during the same year to account for this variation. Although we can't say for sure how big an effect such a variation would have, we think that in a particularly dry year, some relatively small ponds might dry out to the point where the protective effect of the islet would be reduced or eliminated. This could be another advantage of selecting islets surrounded by deeper water.**

246: 3$^{rd}$ and 4$^{th}$ scale = ? I am not familiar with this formulation?

**RESPONSE: The third and fourth scales of selection in wildlife habitat selection represent different levels or scales at which animals make "choices" about habitat use, with the third scale focusing on broader landscape or habitat patch selection and the fourth scale focusing on finer-scale microhabitat selection within those patches. Understanding selection at these multiple scales is essential for comprehensively assessing wildlife habitat preferences and conservation needs.**

**We have added a simple definition to the manuscript:** *"3rd focusing on broader habitat patch selection and 4th scale focusing on finer-scale microhabitat selection within those patches"*.

248: main nest predator: this is questionable, because your birds nest on areas where foxes cannot predate. Any data on nest success and causes of failure for your birds?

**RESPONSE: See Answer to *Methods L97-99* for more details on this topic.**

251: how would the reduced predator abundance come about?

**RESPONSE: To avoid confusion, we removed the statement as it refers only to small islands (not islets).**

252: quality of islets should thus be based on = unclear.

**RESPONSE: We corrected the sentence:** "*The quality of islets in terms of their capacity to reduce predator access should therefore be based on their physical characteristics that can impede predator movements.*"

254 (and elsewhere): references should be ordered from old to new, I believe.

**RESPONSE: We have used the template for formatting citations provided by the magazine.**

261: if you make this statement about maximum jump and leg length, can you also give the exact value of what would be too far or too deep? Otherwise, you should stress that this is a hypothesis.

**RESPONSE: We adjusted the sentence according to your suggestion :** "*Here we hypothesize that the maximum jumping range and leg length of foxes are likely the two main biomechanical constraints limiting their ability to reach an islet without swimming.*"

273: cyclical process: please explain, and what is the duration of one cycle?

**RESPONSE: We added a sentence to clarify this:** "*Permafrost polygons typically form due to the repeated freezing and thawing of the ground surface in polar regions. Degradation of polygons is a cyclical process, driven by the freeze-thaw cycle (French, 2017) ...*"

**Generally, in polar regions where permafrost is prevalent, the freeze-thaw cycle occurs annually, leading to the continuous degradation and reformation of permafrost polygons. However, the specific duration of each cycle and the overall rate of polygon degradation can vary depending on local environmental conditions and the intensity of freeze-thaw processes. We therefore have no precise answer.**

275: you did not show the origin, but assumed/assigned the origin based on visual characteristics. How reliable was that method? (see also L. 134).

**RESPONSE: See above – response to *L134*.**

284: some of them = how many exactly?

**RESPONSE: We tried to identify processes using a qualitative scale, where only the ones we were sure about were used for analysis. As stated in Results 3.2, "***In 68 cases [of 396 known islets], we couldn't attribute a specific process because some islets weren't clearly visible on satellite images, and field observations lacked the detail needed for a single process assignment.***"**

299: should be = is it not possible to test this? And could you say 'were'?

**RESPONSE: We modified the sentence to avoid confusion:** "*The degradation of coastal ridges generated few islets in the landscape and their close parallel organization are more likely to generate islets close to shore, which are less selected by birds.*"

304: various scales = do you mean temporal and spatial? Or different values of those? Please make explicit.

**RESPONSE: We have specified both scales.**

313: contingent upon … areas = this is quite cryptically formulated. Can you reformulate it?

**RESPONSE:  We reformulated as** "*[...] a warming-induced increase in the rate of degradation could further influence the availability of islets, depending on the current extent of degradation observed in wetland areas.*"

321: see general comment about the apparent mixing of long-term and short term (climate change) processes in degradation of polygons.
&
333-334: but how? How does climate change affect this interaction? Temperature goes up, permafrost disappears, polygons may break down, which could first make more islets, but later they may also break down? Furthermore, climate change has also other effects (earlier snowmelt, which increases breeding propensity and success in geese, but potentially has contrasting mismatch effect on chick growth).

*Linking geomorphological processes and wildlife micro-habitat selection: nesting birds select refuges generated by permafrost degradation in the Arctic* (egusphere-2023-2240)

**RESPONSE: It will take millennia for permafrost to disappear from Bylot Island given its thickness of several 100s m. Temperature goes up: near surface permafrost thaws, where ice wedges are present, polygonal ridges may collapse. It is not an even process and segments (sections) of ridges subside and collapse. This creates islets. When permafrost thawing and ice wedge melting continues, the size of the islet's changes, they become smaller. On the other hand, other sections of ridges can be affected, and new islets are formed. Formation and disappearance of islets is a process that occurs on decade to century timescales.**

**Changes in precipitation will also affect the dynamics of islets. Snowfall is redistributed through the landscape by wind and snow cover thickness is essentially a function of surface roughness which is nearly constant or changes very slowly. Therefore, an increase in snowfall should not have a major impact on heat transfer to the permafrost. Simulations of future Arctic climate suggest an increase in rainfall. This phenomenon could potentially significantly alter heat transfer and trigger and speed permafrost degradation and the initiation and evolution of islets. Intense precipitation events generating run-off could trigger thermo-erosion processes which are known to degrade ice-wedge polygons orders of magnitude faster than atmospheric warming. Overall, climate change is expected to initiate permafrost degradation, islet formation and changes in islets topography over time. This implies that islets should be more common in the future.**

329: why mention trait?

**RESPONSE: Trait diversity (also called functional diversity) refers to the variety of organismal traits that influence one or more aspects of the functioning of an ecosystem. It reflects the functional diversity of species, with high trait diversity potentially enhancing ecosystem stability. If the availability of islets was to decrease, species unable to defend themselves and primarily reliant on islets for nesting successfully may decline in abundance. This could potentially lead to the exclusion of these species from the ecosystem, resulting in a decrease in overall ecosystem trait diversity. We clarified the statements in the Discussion:** *"The persistence of vulnerable prey can be strongly affected by predation in the Arctic tundra (Beardsell et al., 2023), and changes in the availability of refuges could affect community trait diversity. Due to their relatively high body and egg size, birds such as Cackling geese, Glaucous gulls and Red-throated loons are likely easy to detect by predators like foxes (Beardsell et al., 2021). However, they do not have the defensive capabilities of larger tundra nesting species, such as Greater Snow geese and Snowy owls (Duchesne et al., 2021). They are mainly found nesting on refuges such as islets and may not be able to persist in a landscape without islets."*

**Appendix A**
"close to shore and in shallow ponds (bottom visible)" = this goes towards circular reasoning. This should be a result of your survey, not a characteristic how to recognise islet types.
**RESPONSE: We clarified in the table.**

**Appendix C**
383: declining: specify function, a negative exponential?
**RESPONSE: We have clarified by adding** *"such as negative exponential functions paired with a distance function".*

**Appendix E**
"occupied by a nesting bird": again, how to deal with multiple birds, or spec A in year 1, spec B in year 2? (see also comment about L. 119-122).
**RESPONSE: See response to** *L119-122.*

**Appendix F**
It is not clear to me what is the added value of this figure compared with Figure 3.
**RESPONSE: The figure shows the classification of processes generating islets for the 396 islets found in the study area, of which only 350 were kept for subsequent analysis (i.e., islets with known DISTANCE and DEPTH). It is meant to show that removing 46 islets did not change the overall proportion of islets in each process category.**

Appendix G
Indicate time frame of this process (and effect of climate change on that?).

**RESPONSE:  Initiation of ice wedge degradation can occur over 1 very warm summer, 2-3 consecutive warm summers or over a warming trend (5-10 years) whereas ice wedge melting and ridge collapse, associated with the development of ponds occur over decades to centuries. We have added this information to the legend of the figure.**

---

## Referee Report (RR1)

Review of the manuscript titled "Linking geomorphological processes and wildlife micro-habitat selection: nesting birds select refuges generated by permafrost degradation in the Arctic",

by Madeleine-Zoé Corbeil-Robitaille et al.

The reviewed manuscript is an important contribution to comprehending the functioning of the Arctic tundra ecosystem amidst rapid climate change. The paper stands as a rare exemplar of adeptly interconnecting geomorphological processes with bird habitat preferences, both of which are accelerating alongside climate change in the Arctic. These changes yield diverse effects on Arctic fauna, occasionally diverging from the anticipated negative impacts. Perhaps the discussion on the ramifications of climate change in the Arctic, particularly the ice loss, on the abundance, distribution and availability of breeding and feeding places of birds is worth expanding. An example is the relatively well-recognized effects of the rapid melting and retreat of tidal glaciers, which create attractive feeding places for birds and marine mammals (see, e.g. https://doi.org/10.1038/srep43999; https://doi.org/10.1007/s10584-016-1853-4; https://doi.org/10.1016/j.jmarsys.2013.09.006).

The authors of the study emphasize the importance of safe refuges for maintaining the population of prey species and, consequently, the diversity of Arctic fauna. In this instance, such refuges take the form of mid-lake islands, which are inaccessible to foxes. This situation brings to mind the anthropogenic refuges utilized by certain other Arctic birds. For example, the extensive eider colony in Longyearbyen, Spitsbergen, is situated just beyond the confines of the large barking city kennel. No fox goes there.

The studies were appropriately designed and implemented, and the results were analyzed using advanced statistical procedures and models. I believe that the manuscript meets the criteria outlined by "Biogeosciences" in terms of both content and formal relevance. Therefore, I recommend its approval and publication.

Below are some my comments and suggestions.

- In my opinion, there is a lack of information on the number of foxes and the predation pressure they exert on birds in the study area. The authors assumed that the pressure is so significant that it forces birds to select safer nesting places. However, if there are few foxes and their pressure is negligible, the conclusions from the work lose some weight.
- Are there seasonal fluctuations in the water level and therefore the area of ponds and islands, which would mean changes in their availability for foxes, and have they been taken into account?
- The description of the study area refers to lowland and coastal areas. Were there any differences found in the colonization of ponds by birds between these two areas?
- It is difficult to spot Bylot Island on the attached map. It is indeed black, but small enough that an arrow would certainly be useful.
- The distance and abundance of feeding grounds used by birds also influence the selection of nesting sites by birds. This has different meanings for gulls, loons and geese. The latter graze on tundra vegetation, usually near nesting sites. Sometimes they can also move with their chicks to more abundant pastures, but they always need to have a body of water nearby to escape from predators. I don't know what the situation is like in Bylot I. and whether it may be important for the habitat preferences of Cackling Geese.

---

## Referee Report (RR2)

REVIEW by Dr. Donald Reid

(Wildlife Conservation Society Canada)

OF

"Linking geomorphological processes and wildlife micro-habitat selection: nesting birds select refuges generated by permafrost degradation in the Arctic."

 Authors: Madeleine-Zoé Corbeil-Robitaille, Éliane Duchesne, Daniel Fortier, Christophe Kinnard, Joël Bêty

**Summary Comments**

This is a very interesting and valuable, field-based, study of an influential process in Arctic ecology – the degradation of polygonal terrain and associated shifts in availability and use of islets as predation refuges. It is certainly worthy of publication.

Overall, my suggestions for changes have to do with (a) relating this study more completely to both the broader Arctic context and the full degradational continuum for Arctic water bodies which can include talik drainage of lakes and ponds, (b) more careful use of technical terms on some points, (c) providing a bit more detail on some processes to illustrate some key points (the biologically-trained readership would benefit from more detail on the  geomorphological and hydrological processes), and (d) minor improvements to the English.

**Comments regarding content and interpretation**

96      Are these measures of active layer depth just referring to upland tundra without standing water? If so, then that is worth mentioning. Are there any data on active layer depth under water bodies? What happens to the water in these ponds in winter (does it always completely freeze to the bottom)?

101      What does "essentially" mean? A better description in English could be found to indicate frequency of nesting on islets as compared to nesting on "mainland" ("the great majority"?)

110      It would be useful to provide the size (square kilometres) of the "study area", and the total number of individual water bodies, so that readers can understand the density of islets and ponds in this ecosystem, for comparison to other Arctic ecosystems.

Table 1 (i) In line 3, reference is made to "glacial drift". Glacial drift is defined as a set of materials left behind by glaciers and their meltwater; it is not a process as such. So the "boulders" are not deposited "by" drift, but "as a component of" drift. (ii) In line 5, "exposition" is not a very appropriate word in English for this process (it generally refers to abstract ideas); "exposure" is used to refer to physical processes that "reveal" something new and would be a better term to use.

Table 2 Please explain (in the caption) why some of the parameters are displayed in bold font in the right-hand panel.

Figure 4        The Caption text includes: "The islets used by nesting birds are shown using dark filled circles." This seems insufficient to fully explain these circles: (a) mention that there is an interpretive legend (bottom right) would be useful); (b) Is each number in the legend equal to the total number of islets used by nesting birds of each species?; (c) there are four sizes of circle, each with a unique whole integer value in the legend (categorical summary): how are numbers of islets in between these integer values to be inferred (are the four values just maximums higher than the next lowest category)?

285    I don't think this sentence is technically correct, nor intended. Cryoturbation and frost-cracking primarily affect the *ground surface* (and in winter) not the *surface of the permafrost.* The surface of the permafrost is at the base of the active layer, and varies from year to year depending on how deeply the thaw penetrates the ground and how much meltwater pools at the base of the active layer.

287    Mention is made of "permafrost polygons". Yes, the polygonal patterns are related to permafrost presence, but the polygons themselves are not comprised of permafrost (except for the ice wedge parts), so using permafrost as an adjective here seems inappropriate. The sentence would be correct if "Permafrost" is dropped from the beginning, and the text explains that the repeated annual freezing and thawing of water is supported over many years by the inability of that water, in the active layer, to drain away because of the underlying permafrost.

325    The last sentence of this paragraph talks about potential processes leading to change in ponds and islets. I think this needs more complete discussion. (A) It seems to me (based on your information and other literature) that the process of degradation of polygonal terrain to form thermokarst ponds is already well underway; this already defines the distribution and size of ponds to some extent so is not just a future outcome. In addition, what do the authors know about any efforts to quantify this historical pattern based on sequences of satellite images? If this has not been done, then it seems like a useful exercise to attempt, and worth mention in this discussion. (B) Mention is made of "positive feedback" but it is unclear how this would work. Please explain. For example, does this have to do with rising temperature of the water, or increasing wind fetch on bigger ponds? The first is an amplification of the existing direct effect, and the second might be considered a positive feedback depending on how frequent and certain the coalescence of ponds is over time. Wind deserves discussion because it is a dominant feature of Arctic weather, and could affect islet longevity and turnover through the erosive power of waves. (C) Earlier in the paragraph "hydrological flux" is mentioned, without explanation as to specific process. Are there demonstrated trends in snowfall in this region that are affecting the amount of surface water after melt? This possibility needs discussion, because of immediate within-year effects on islet availability. In fact, the amount of surface water is probably the dominant factor, in conjunction with air temperature, driving degradational processes and suitability of islets for nesting.

337    The sentence starting "The lowlands,…" would benefit from more discussion to place it both in the larger Arctic context and in the complete degradational continuum. This study is in the High Arctic. Degradational effects are occurring in this study area, so how does that relate to other Arctic regions, especially the Low Arctic? For example, there is a further step in the degradational continuum wherein ponds/lakes disappear, because of draining through a talik in the underlying permafrost which itself has degraded deeply. This is well documented in subarctic areas. I presume you have no evidence of this yet from Bylot Island, but what is the risk of it occurring (i.e. how deep is the permafrost underlying the study area)? Do you know of any evidence of draining of thermokarst lakes in the Low Arctic (e.g., Chen et al. 2022 Sci Tot Env, https://doi.org/10.1016/j.scitotenv.2021.150828. )?

339	The text suggests "Additional research". Please suggest some specifics. For example, can sequences of remotely-sensed imagery provide inference about these processes and their possible trends, and also the fate of islets themselves? What about forward-looking monitoring, field-based and/or with remote sensing?

**Editorial Points**

37	The sentence is incomplete: "…diversity of biological ? "

55	The reference to Caro 2005 is incorrectly written; there is no need for first name (Caro, 2005)

62	The sentence is incomplete: "…surrounding dryer ? "

95	I think the valleys are created by the rivers, so "…valleys with glacial rivers…" would be better

96	The parentheses before the word "active" need to be closed after "(Fortier et al. 2006)"

97	The second use of the word "wetlands" should be "wetland"

286	"reliefs" should be "relief"

311	"drifts" would better be "drift"

317	Title of section 4.4. "refuges sensitivity" needs to change (in terms of suitable English). This section is mostly about the *longevity of islets*, or *susceptibility of islets to change.*

334	The word "together" is unnecessary and redundant.